# PD-L1 degradation is regulated by electrostatic membrane association of its cytoplasmic domain

Maorong Wen [1,7✉], Yunlei Cao[1,2,7], Bin Wu[3,7], Taoran Xiao[1,2], Ruiyu Cao[1,2], Qian Wang[1,2], Xiwei Liu[1,2], Hongjuan Xue[3], Yang Yu[3], Jialing Lin [4,5], Chenqi Xu [1,2], Jie Xu[6] & Bo OuYang [1,2✉]

The cytoplasmic domain of PD-L1 (PD-L1-CD) regulates PD-L1 degradation and stability through various mechanism, making it an attractive target for blocking PD-L1-related cancer signaling. Here, by using NMR and biochemical techniques we find that the membrane association of PD-L1-CD is mediated by electrostatic interactions between acidic phospholipids and basic residues in the N-terminal region. The absence of the acidic phospholipids and replacement of the basic residues with acidic residues abolish the membrane association. Moreover, the basic-to-acidic mutations also decrease the cellular abundance of PD-L1, implicating that the electrostatic interaction with the plasma membrane mediates the cellular levels of PD-L1. Interestingly, distinct from its reported function as an activator of AMPK in tumor cells, the type 2 diabetes drug metformin enhances the membrane dissociation of PD-L1-CD by disrupting the electrostatic interaction, thereby decreasing the cellular abundance of PD-L1. Collectively, our study reveals an unusual regulatory mechanism that controls the PD-L1 level in tumor cells, suggesting an alternative strategy to improve the efficacy of PD-L1-related immunotherapies.

[1] State Key Laboratory of Molecular Biology, Shanghai Institute of Biochemistry and Cell Biology, Center for Excellence in Molecular Cell Science, Chinese Academy of Sciences, Shanghai, China. [2] University of Chinese Academy of Sciences, Beijing, China. [3] National Facility for Protein Science in Shanghai, ZhangJiang lab, Shanghai Advanced Research Institute, Chinese Academy of Sciences, Shanghai, China. [4] Department of Biochemistry and Molecular Biology, University of Oklahoma Health Sciences Center, Oklahoma City, OK, USA. [5] Stephenson Cancer Center, Oklahoma City, OK, USA. [6] Institutes of Biomedical Sciences, Fudan University, Shanghai, China. [7] These authors contributed equally: Maorong Wen, Yunlei Cao, Bin Wu. ✉email: mrwen@sibcb.ac.cn; ouyang@sibcb.ac.cn

The important discovery that programmed death ligand 1 (PD-L1) expression on tumor cells inhibits programmed cell death 1 (PD-1) on T cells to escape immune surveillance has opened an era of tumor immunotherapy[1–4]. PD-L1 is a type I transmembrane protein[5] containing an extracellular domain (ECD), a transmembrane domain (TMD), and a cytoplasmic domain (CD). It is highly expressed on the surface of many cancer cells[6,7]. Early functional and structural studies have revealed that the canonical immunoglobulin (Ig)-like ECD of PD-L1 binds to PD-1 on T cells to inhibit their tumor-killing activity[5,8]. The PD-L1-ECD-targeting drugs that were developed to recover the tumor-killing activity of T cells have achieved great clinical success in treating a wide range of cancers[9,10]. However, the therapeutic resistance to these anti-PD-L1-ECD drugs has been increasingly observed in patients[11–13], highlighting the need for alternative strategies to overcome the acquired resistance.

The small cytoplasmic domain of PD-L1 (PD-L1-CD; containing residues 260–290) is involved in multiple regulation pathways controlling PD-L1 protein stability and degradation, and hence a potential new target inside the cell for the immune surveillance. PD-L1-CD was reported to bind the cullin 3$^{SPOP}$ E3 ligase (SPOP) that decreases the cellular abundance of PD-L1 through ubiquitination-dependent degradation[14]. Deletion of the cytoplasmic tail disrupts the binding to SPOP and renders PD-L1 resistance to the ubiquitination-dependent proteasome-mediated degradation. Xu lab revealed that PD-L1 undergoes lysosome-dependent proteolysis after huntingtin interacting protein 1 related (HIP1R) binds to its C-terminus and delivers it to the lysosome[15]. Xu and Hung lab separately demonstrated that palmitoylation of C272 in the PD-L1-CD regulates PD-L1 stability and trafficking[16,17]. Furthermore, PD-L1 was reported to contain a conserved class of sequence motifs in PD-L1-CD that mediate crosstalk with interferon signaling[18]. Besides the multiple regulatory roles of the cytoplasmic domain, several cancer-derived mutations were found within this domain, including R260C, R262K, D276Y, and T290M, alluding its critical role in cancer cell survival[18]. Interestingly, the peptides designed to target PD-L1-CD reduced the PD-L1 protein levels by enhancing the lysosomal degradation[15] or blocking the palmitoylation[16], resulting in reduced binding to PD-1 on T cells, thereby enhancing their antitumor activity. Therefore, targeting of PD-L1-CD within the cancer cell is a promising route for the development of anticancer immunotherapy[19,20].

Despite of the high biological relevance of PD-L1-CD in regulating PD-L1 levels in tumor cells, the molecular recognition and interaction of PD–L1–CD that control PD-L1 levels remain a mystery. Previous studies showed that PD-L1 is highly expressed on the plasma membrane in tumor cells[6,7], of which the inner leaflet is enriched in acidic phospholipids, including phosphatidylserine (PS) and phosphatidylinositol (PI). The asymmetric distribution of lipids results in negative charges on the cytoplasmic face of the plasma membrane[21]. Moreover, recent studies revealed that PD-L1 is not only located on the cell membrane, but also in extracellular vesicles, such as exosomes, which facilitates tumor cells to evade immune surveillance[12,13,22]. Exosomes are also enriched in acidic lipids such as PS[23], indicating that PD-L1 has a preference to locate in an acidic lipid-rich environment. The palmitoylation of C272 stabilizes the membrane insertion of PD-L1-CD and thus suppresses PD-L1 degradation[17], suggesting that the membrane association of PD-L1-CD has a significant effect on stabilizing PD-L1. More broadly, the asymmetric lipid distribution with excess negative charges on the plasma membrane appears to regulate the signaling via several type I transmembrane proteins, such as CD3ε/ζ and CD28 in T cells[24–26].

Therefore, in this study we performed structural and functional investigations of the lipid regulation of PD-L1-CD. We found that the membrane insertion of PD-L1-CD is regulated by negatively charged acidic phospholipids. The electrostatic interactions between acidic phospholipids and basic residues in PD-L1-CD proximal to the TM domain are critical for the membrane association. Using nuclear magnetic resonance (NMR) technology, we determined how PD-L1-CD binds to a lipid bicelle, in which three arginine residues are inserted into the lipid bilayer. Sequestration of these arginines into the plasma membrane enhanced the PD-L1 degradation in cells. Interestingly, we observed that metformin can disrupt the interaction between PD-L1-CD and membranes. Further, mutagenesis and cell-based functional assay showed that disruption of the membrane association by metformin reduces PD-L1 levels. And, the PD-L1-CD-membrane interaction controls the abundance of PD-L1 in tumor cells and thereby regulates the immunosurveillance. Thus, our study uncovers a molecular mechanism that regulates PD-L1 levels, providing insights into PD-L1-mediated immune evasion of tumor cells that can be potentially targeted by drugs.

## Results

**Acidic lipids regulate membrane interaction of PD-L1-CD.** To explore whether acidic lipids regulate PD-L1-CD protein–membrane interactions, we expressed the residues 260–290 of human PD-L1 (CD$_{260-290}$) (Supplementary Fig. 1a) in *E. Coli* that was fused to the C-terminus of a His$_8$-SUMO sequence with a PreScission protease (3 C) cleavage site in between. The fusion protein was purified by nickel affinity chromatography followed by 3 C cleavage and reverse-phase HPLC to remove the His$_8$-SUMO tag (Supplementary Fig. 1b–d).

The purified CD$_{260-290}$ protein was incubated with 1,2-dimyristoyl-sn-glycero-3-phosphocholine (DMPC) liposomes supplemented with 25% 1,2-dimyristoyl-sn-glycero-3-phosphoethanolamine (DMPE), 1,2-dimyristoyl-sn-glycero-3-phosphoglycerol (DMPG), 1,2-dimyristoyl-sn-glycero-3-phospho-L-serine (DMPS), or 5 or 15% cardiolipin (CL), respectively. After an ultracentrifugation, CD$_{260-290}$ was found in the pellet associated with the liposomes containing the acidic phospholipid DMPG, DMPS or CL but not the zwitterionic phospholipid DMPE and/or DMPC (Fig. 1a). These results indicate that the acidic phospholipids are required for the membrane binding of CD$_{260-290}$.

We further used solution NMR spectroscopy to study the PD-L1-CD-membrane interaction as previously reported[24]. Isotope-labeled CD$_{260-290}$ was reconstituted into the zwitterionic DMPC/1,2-dihexanoyl-sn-glycero-3-phosphocholine (DH$^6$PC) bicelles (molar ratio q = 0.8) containing DMPG at different molar ratios (Supplementary Fig. 2a). Two-dimensional (2D) $^1$H-$^{15}$N transverse relaxation optimized spectroscopy (TROSY) experiments were used to monitor the chemical shift and intensity perturbations. A continuous fashion during the titration was observed on the N-terminal residues of CD$_{260-290}$ (Supplementary Fig. 2a-c), i.e., a single resonance peak for each perturbed resonance of CD$_{260-290}$ at any titration point, indicating that CD$_{260-290}$ is in fast exchange on the chemical shift time scale between membrane-bound and unbound forms. We noticed that CD$_{260-290}$ in DMPC/DH$^6$PC bicelles (molar ratio q = 0.8) did not have detectable chemical shift differences from the protein in solution (Fig. 1b and Supplementary Fig. 2d), while CD$_{260-290}$ in acidic DMPG/DH$^6$PC bicelles (q = 0.8) displayed significant chemical shift and peak intensity changes in most N-terminal resonance peaks (Fig. 1b and Supplementary Fig. 2d), suggesting that CD$_{260-290}$ specifically binds to acidic phospholipids in the bicelles. Moreover, addition of CD$_{260-290}$ to 1-palmitoyl-2-oleoyl-sn-glycero-3-(phospho-rac-(1-glycerol)) (POPG) /DH$^6$PC bicelles (q = 0.8) induced chemical shift changes similar to those induced by DMPG/DH$^6$PC bicelles (Supplementary Fig. 2e).

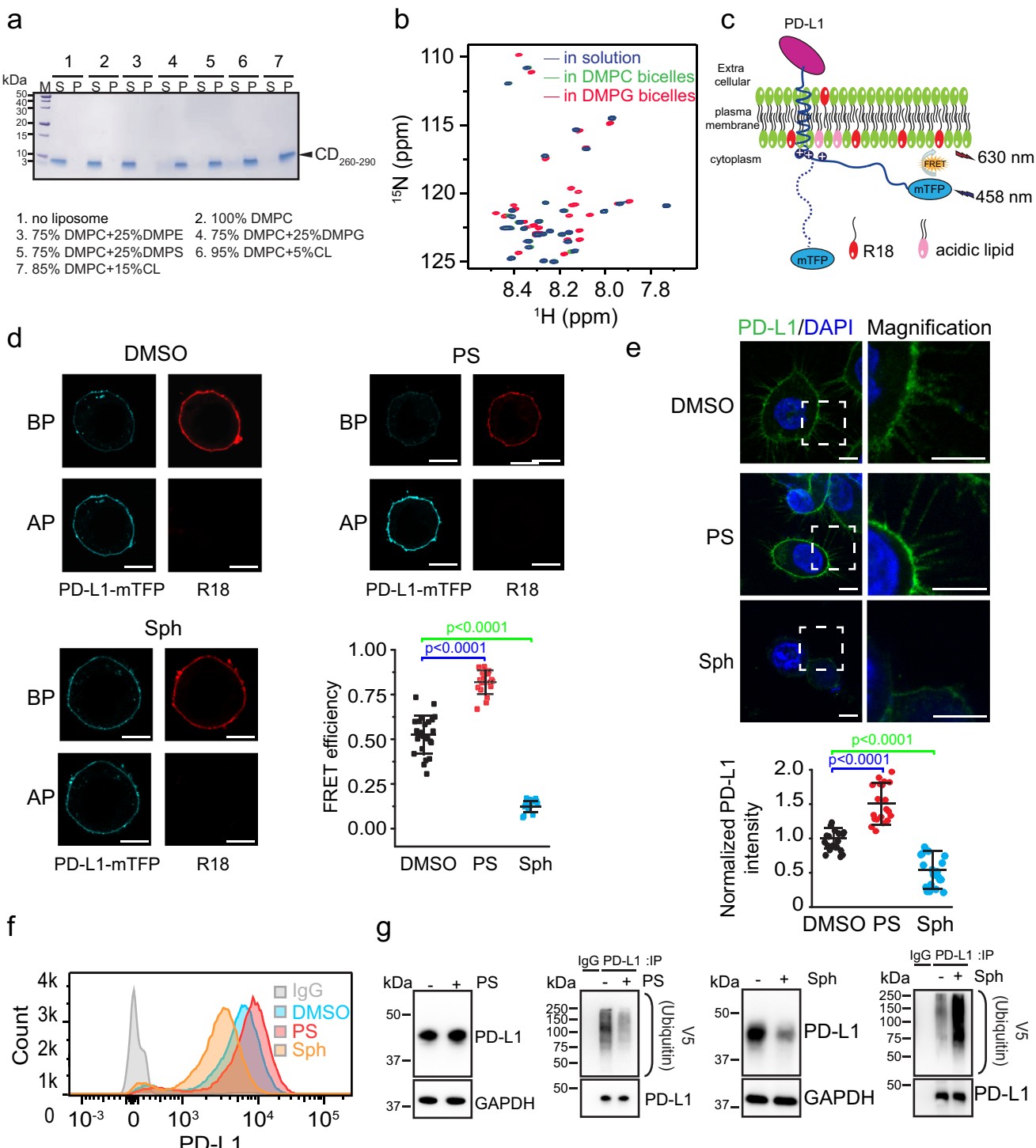

To test the effect of acidic lipids in live cells, we used fluorescence resonance energy transfer (FRET) to monitor the membrane binding of PD-L1-CD following an established protocol[24,26], in which the monomeric teal fluorescent protein (mTFP) was the donor and the octadecyl rhodamine B (R18) was the acceptor (Fig. 1c). The mTFP was fused to the C-terminus of full length PD-L1 (PD-L1-mTFP). The PD-L1-mTFP fusion protein was stably expressed in RKO cells whose plasma membranes were labeled with the R18 dye. The FRET efficiency between PD-L1-mTFP and the R18 dyes in the membranes was first measured using the donor dequenching method. The result showed that after a treatment of the cells with PS, which increases

the content of acidic lipids in the plasma membrane[27], a higher FRET efficiency was detected compared to the solvent-treated cells, indicating stronger interactions between PD-L1-CD and the membrane with more acidic phospholipids (Fig. 1d). This result was further confirmed by the FRET efficiency determined using the donor quenching approach (Supplementary Fig. 3). In addition, a lower FRET efficiency was obtained when the RKO cells were treated with sphingosine (Sph, a membrane-permeant base)[27], as expected for the reduced negativity of the plasma membrane by Sph (Fig. 1d).

The effects of PS and Sph on the endogenous PD-L1 level on the RKO cell surface were further examined by confocal imaging

**Fig. 1 Acidic phospholipids are required for recruitment of PD-L-CD into model and native membranes. a** $CD_{260-290}$ was incubated without or with DMPC liposomes supplemented with DMPE, DMPG, DMPS, or CL as indicated on the bottom. The fractions from the pellet (P) and the supernatant (S) were separated by ultracentrifugation and analyzed by SDS-PAGE. Two independent experiments were performed with similar results. **b** Superimposed $^1$H-$^{15}$N TROSY-HSQC spectra of $CD_{260-290}$ in a solution buffer (25 mM MES pH 6.5) (blue) and in the same solution plus DMPC/DH$^6$PC (green) or DMPG/DH$^6$PC (red) bicelles. The results are representative of three independent experiments. **c** Schematic illustration of the FRET assay for measuring the interaction between mTFP (blue) fused to the C-terminus of PD-L1 and R18 dye (red) incorporated into the plasma membrane. The membrane contains acidic (pink) and other (green) lipids. The basic patch in PD-L1-CD is indicated by $+$ symbols. **d** FRET detected lipid effects on PD-L1-CD-membrane interaction. Fluorescent images of the cells treated with DMSO, 5 μM PS or 2 μM Sph were taken to detect the mTFP or R18 dye emission before or after photobleaching (BP or AP) the R18 dye, respectively. Scale bars = 10 μm. The FRET statistical results from cells treated with DMSO ($n = 24$), PS ($n = 18$) or Sph ($n = 17$) are shown as the mean ± standard deviation (SD). Statistical differences were determined by unpaired two-sided Student's t-test. **e** Immunofluorescence detection of endogenous PD-L1 in RKO cells treated with DMSO, 5 μM PS or 2 μM Sph. Scale bars = 5 μm in all panels. The fluorescence intensities from the cells were quantified and shown as mean ± SD from $n = 20$ cells under each treatment. P values were determined by two-sided Student's t-test. **f** Cell surface levels of PD-L1 in RKO cells analyzed by flow cytometry after treatment with DMSO, 5 μM PS or 2 μM Sph for 6 h. **g** PD-L1 level (left) and ubiquitination (right) in RKO cells analyzed by western blotting. RKO cells were cultured for 12 h with or without 5 μM PS or 2 μM Sph. Two independent experiments were performed with similar results. Source data are provided as a Source Data file.

(Fig. 1e) and flow cytometry (Fig. 1f). The results showed that the addition of PS to the culture medium significantly upregulated, whereas the addition of Sph significantly downregulated, the cell-surface expression of PD-L1. Consistently, western blot analysis showed that the PD-L1 level increased by the PS treatment but decreased by the Sph treatment (Fig. 1g). In accordance, the ubiquitination of endogenous PD-L1 that was substantially decreased after the PS treatment increased after the Sph treatment (Fig. 1g). Together, the live-cell results show that changing the net charge of the plasma membrane specifically influences the interaction of PD-L1-CD with the membrane as expected from the NMR and biochemical data. And, the membrane interaction in turn controls the cellular abundance PD-L1.

**Membrane insertion of PD-L1-CD**. We next studied how $CD_{260-290}$ interacts with the membrane using solution NMR spectroscopy. As mentioned above, a series of NMR spectra at different ratios of DMPG/DMPC showed that the chemical shifts of the N-terminal resonances moved continuously in straight lines (Supplementary Fig. 2a), indicating that $CD_{260-290}$ exists a two-state shift from membrane unbound to bound. To trap $CD_{260-290}$ in a membrane-bound state, we chose 100% DMPG/DH$^6$PC or 100% POPG/DH$^6$PC for the further investigation on protein–membrane interaction. The $CD_{260-290}$ protein was first reconstituted in DMPG/DH$^6$PC bicelles (q = 0.8). Essentially complete backbone and side-chain resonance assignments were accomplished through a standard set of triple resonance experiments (Fig. 2a). The intramolecular NOE restraints showed that the cytoplasmic domain are largely dynamic with no NOEs characteristic of secondary structures.

The membrane-bound protein was then characterized using the protein–lipid intermolecular NOEs measured by the protein–lipid NOESY experiments ($\tau_{NOE} = 200$ ms) (Fig. 2b), in which $CD_{260-290}$ was completely deuterated to eliminate any signals from the backbone amides to side chains and reconstituted in the bicelles containing the DH$^6$PC with deuterated acyl chains. 100% POPG/DH$^6$PC bicelles were used in this experiment to keep the lipid acyl chains with the same length as the native membranes. Three arginine residues (R260/R262/R265) showed strong NOEs (distance of < 5 Å) to lipid acyl chains and headgroups (Fig. 2b, c), suggesting that these residues adopt an orientation toward the interior of the lipid bilayer. Such an orientation can facilitate the ionic interactions between the basic residue side chains and the phosphate groups in the lipid headgroups. V269 showed strong NOE signal to the methyl but weak signal to the methylene of lipid acyl chains, suggesting it is deeply inserted into the lipid bilayer (Fig. 2b).

In addition to NOE-based determination of $CD_{260-290}$ membrane insertion, we performed paramagnetic relaxation enhancement analysis for qualitative validation of the membrane-embedded $CD_{260-290}$ conformation. Two paramagnetic relaxation enhancement (PRE) probes were used to measure residue-specific depth immersion of the protein in the lipid bilayer region of the bicelles, as previously described[28] (Supplementary Fig. 4a). The residue-specific PRE amplitudes (PRE$_{amp}$) from the water-soluble paramagnetic probe Gd-DOTA indicate that the N-terminal half of $CD_{260-290}$ is mostly inserted into the lipid bilayer with residues 265–270 displaying the lowest PRE$_{amp}$ inferring their membrane partition, while the C-terminal half is exposed to water with high PRE$_{amp}$ (Supplementary Fig. 4b). The results from the lipophilic PRE probe 16-DSA (Supplementary Fig. 4c) showed a clear cutoff of PRE$_{amp}$ between the membrane-embedded residues 260–274 and the water exposed residues 275–290 (Supplementary Fig. 4d). In short, the PRE data from the paramagnetic probe titrations are consistent with the protein–lipid NOE data, strengthen a model for partially membrane-embedded $CD_{260-290}$.

**Key arginine residues in PD-L1-CD are critical to the membrane interaction**. In the membrane-bound $CD_{260-290}$, several basic residues in the N-terminal half of $CD_{260-290}$, including R260, R262, K263, R265, K270, and K271, may interact with the lipid headgroups. Sequence analysis of PD-L1-CD shows that this basic region in the juxtamembrane segment is evolutionarily conserved between species, though the exact positions of the basic residues are not well conserved (Supplementary Fig. 5a, b).

The 3D NOESY spectra showed that R260, R262 and R265 have the strongest NOE signals to lipid headgroups and acyl chains, whereas K270 and K271 have weak NOE signals to lipids (Fig. 2b). We made R260E/R262E/R265E ($CD_{260-290}^{3RE}$) and K270E/K271E ($CD_{260-290}^{2KE}$) mutants to determine the contribution of these positively charged residues to the membrane interaction. The 2D TROSY spectra showed that the addition of $CD_{260-290}^{3RE}$ mutant to the DMPG/DH$^6$PC bicelles did not induce the resonance shifts that was induced by the wild type (WT) protein ($CD_{260-290}^{WT}$) (Figs. 1b and 3a). In contrast, obvious chemical shift perturbations were observed in the spectra of $CD_{260-290}^{2KE}$ mutant upon the addition of the bicelles (Supplementary Fig. 5c). These spectral data indicate that the $CD_{260-290}^{3RE}$ mutant loses the capacity to bind the membrane even it contains the acidic lipids, but the $CD_{260-290}^{2KE}$ mutant retains the membrane binding capacity. This conclusion was further supported by the liposome binding assay, in which only $CD_{260-290}^{2KE}$ mutant was detected in the DMPG liposome pellet after both mutants were incubated with the liposomes (Fig. 3b and Supplementary Fig. 5d). As a negative control, the R260K/R262K/R265K ($CD_{260-290}^{3RK}$) mutant interact

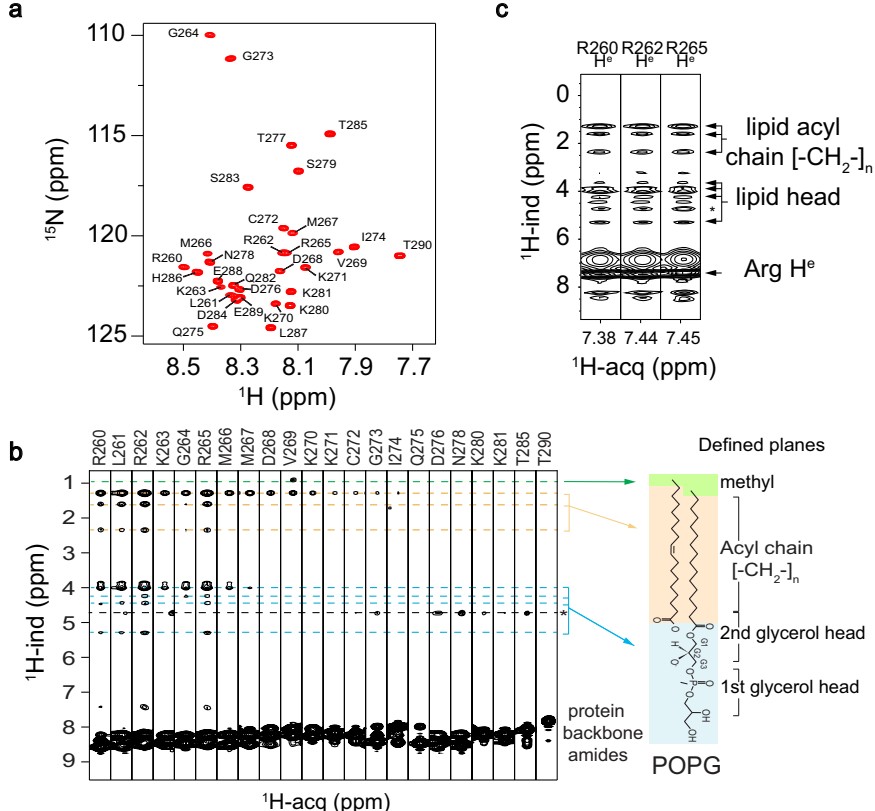

**Fig. 2 The NMR characterization of membrane-bound CD$_{260-290}$. a** $^1$H-$^{15}$N TROSY-HSQC spectrum of CD$_{260-290}$ in POPG/DH$^6$PC bicelles with backbone resonances assigned. The spectrum was recorded at $^1$H frequency of 600 MHz using [$^{15}$N, $^2$H]-labeled protein. **b** Lipid NOEs from CD$_{260-290}$ in POPG/DH$^6$PC bicelles. Strips from 3D $^{15}$N-edited NOESY-TROSY-HSQC ($\tau_{NOE} = 200$ ms) spectra recorded using 0.8 mM [$^{15}$N, $^2$H]-labeled CD$_{260-290}$ in the bicelles with protonated POPG and deuterated DH$^6$PC show the NOEs exclusively between backbone amide protons of CD$_{260-290}$ and methyl protons or methylene protons of lipid acyl chains of POPG, and the NOEs between backbone amide protons of CD$_{260-290}$ and glycerol protons of lipid head groups and water protons (*), mostly for the residues in the N-terminal half of the protein. The spectrum was recorded at $^1$H frequency of 900 MHz. **c** Strips from the same experiment mentioned above showed the NOEs from the side chains of three arginine residues to the lipid acyl chains and headgroups of POPG.

with the lipid bicelles similar to the WT protein as detected by the NMR spectroscopy (Supplementary Fig. 5e) and the liposome binding assay (Supplementary Fig. 5f), implying that the positive charges at these positions instead of the specific residues dictate the ionic protein–lipid interaction.

To validate the role of the positively charged residues in the membrane binding we defined in vitro, we expressed wildtype CD$_{260-290}$ with a C-terminal GFP tag (CD$_{260-290}$$^{WT}$-GFP) in HEK293T cells, and determined the subcellular localization by confocal imaging. CD$_{260-290}$$^{WT}$-GFP is mainly localized on the plasma membrane (Fig. 3c), while CD$_{260-290}$$^{3RE}$-GFP mutant is dispersed in the cytoplasm (Fig. 3d). These results were further confirmed by the FRET measurement with the mTFP-tagged full-length PD-L1$^{3RE}$ mutant (PD-L1$^{3RE}$-mTFP) expressed in the RKO cells showing a lower FRET efficiency to the R18 dyes in the plasma membrane than the full-length wildtype PD-L1 (PD-L1$^{WT}$-mTFP) in the absence of PS (Fig. 3e), while the presence of PS didn't affect the FRET efficiency on PD-L1$^{3RE}$ (Fig. 3e). These mutagenesis data demonstrate that PD-L1-CD binds to the negatively charged membrane via the ionic interaction of the three positively charged juxtamembrane arginine residues, replacement of which with the negatively charged glutamic acid residues can potently disrupt the membrane binding.

**PD-L1-CD-membrane interaction regulates PD-L1 stability.** As we mentioned above, PD-L1-CD regulates PD-L1 stability and

degradation[14,15], and stabilizing the PD-L1-CD-membrane association by palmitoylation[16] suppresses the degradation of PD-L1. To study whether the membrane interaction of PD-L1-CD facilitated by acidic lipids plays a role in the stability of PD-L1, we examined the level of full-length PD-L1$^{WT}$ and PD-L1$^{3RE}$ mutant expressed in HEK293T cells. The cycloheximide (CHX)-chase assay[29] revealed accelerated reduction of PD-L1$^{3RE}$ level than PD-L1$^{WT}$ level (Fig. 4a–c). Cell-surface expression of PD-L1 monitored by flow cytometry showed a more substantial reduction of PD-L1$^{3RE}$ mutant compared to the wild-type protein (Fig. 4d). MG132 (carbobenzoxyl-L-leucyl-L-leucyl-L-leucine), an inhibitor that can effectively block the proteosome activity[30], was introduced to the HEK293T cells expressing PD-L1$^{WT}$ or PD-L1$^{3RE}$ mutant to inhibit their degradation. Western blot analysis indicated that MG132 restored the level of both proteins in HEK293T cells with a more significant increase observed for PD-L1$^{3RE}$ mutant (Fig. 4e).

To demonstrate that the proteasome mediated PD-L1 degradation involves ubiquitination, we determined the level of the cellular ubiquitination of PD-L1$^{3RE}$ mutant relative to PD-L1$^{WT}$. The HEK293T cells expressing PD-L1$^{WT}$ or PD-L1$^{3RE}$ were first treated with MG132, then subjected to PD-L1 immunoprecipitation (IP), followed by immunoblotting analysis with anti-V5 antibody to detect the ubiquitination of full length PD-L1$^{WT}$ and PD-L1$^{3RE}$ mutant. We found more ubiquitination for PD-L1$^{3RE}$ mutant in the presence of MG132 compared to the wild-type protein (Fig. 4f), indicating that the mutant undergoes

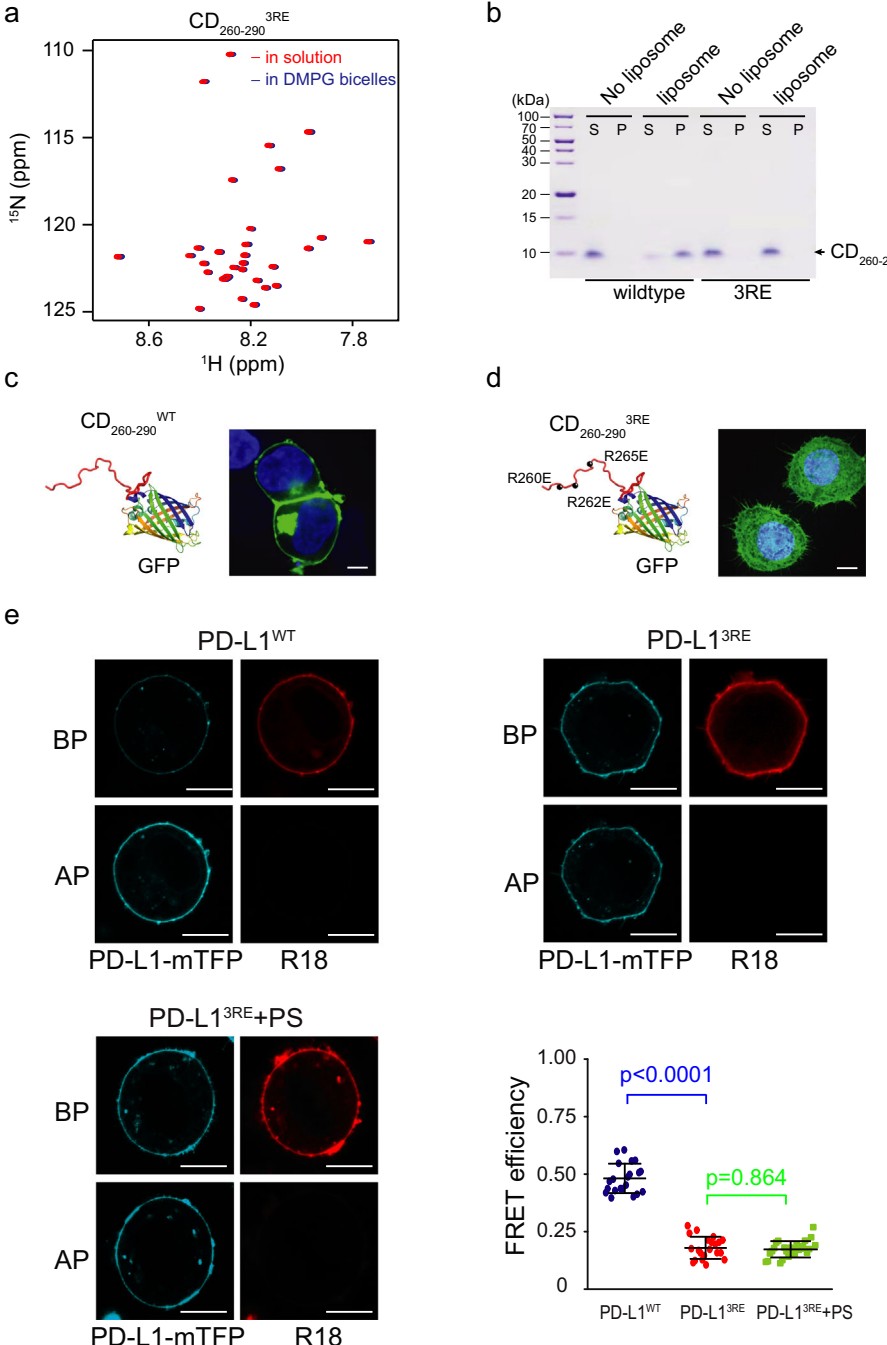

**Fig. 3 Polybasic residues mediate the membrane interaction of PD-L1-CD in vitro and in cells. a** Superimposed 2D $^1$H-$^{15}$N TROSY-HSQC spectra of CD$_{260-290}$$^{3RE}$ mutant in solution (red) and with DMPG/DH$_6$PC bicelles (blue). The spectra were recorded at $^1$H frequency of 600 MHz using [$^{15}$N, $^1$H]-labeled protein. **b** Wildtype and mutant CD$_{260-290}$ association with liposomes ($n = 3$ independent experiments). The liposome binding assay was performed with (labeled as "liposome") or without liposomes (labeled as "No liposome") as in Fig. 1a but for CD$_{260-290}$$^{WT}$ and CD$_{260-290}$$^{3RE}$ mutant. The fractions after centrifugation were analyzed by SDS-PAGE, in which "S" represents the supernatant and "P" represents the pellet. **c**, **d** Localization of wildtype and mutant CD$_{260-290}$ in HEK293T cells. Confocal microscopy images of HEK293T cells expressing CD$_{260-290}$$^{WT}$-GFP (**c**) or CD$_{260-290}$$^{3RE}$ (**d**) protein are shown. The experiment was repeated twice with similar results. Scale bars, 5 μm. **e** Plasma membrane interaction of wildtype and mutant PD-L1 detected by FRET. The FRET efficiency was determined for RKO cells expressing PD-L1$^{WT}$-mTFP (left top), PD-L1$^{3RE}$-mTFP (right top) and PD-L1$^{3RE}$-mTFP treated with 5 μM PS (left bottom) for 2 h. Scale bars, 10 μm. A similar whisker-dot plot shows the statistical results from $n = 24$, 22, or 25 WT or 3RE mutant-expressing cells, respectively (right bottom). Data are represented as mean ± SD. P values were determined by unpaired two-sided Student's t-test. Source data are provided as a Source Data file.

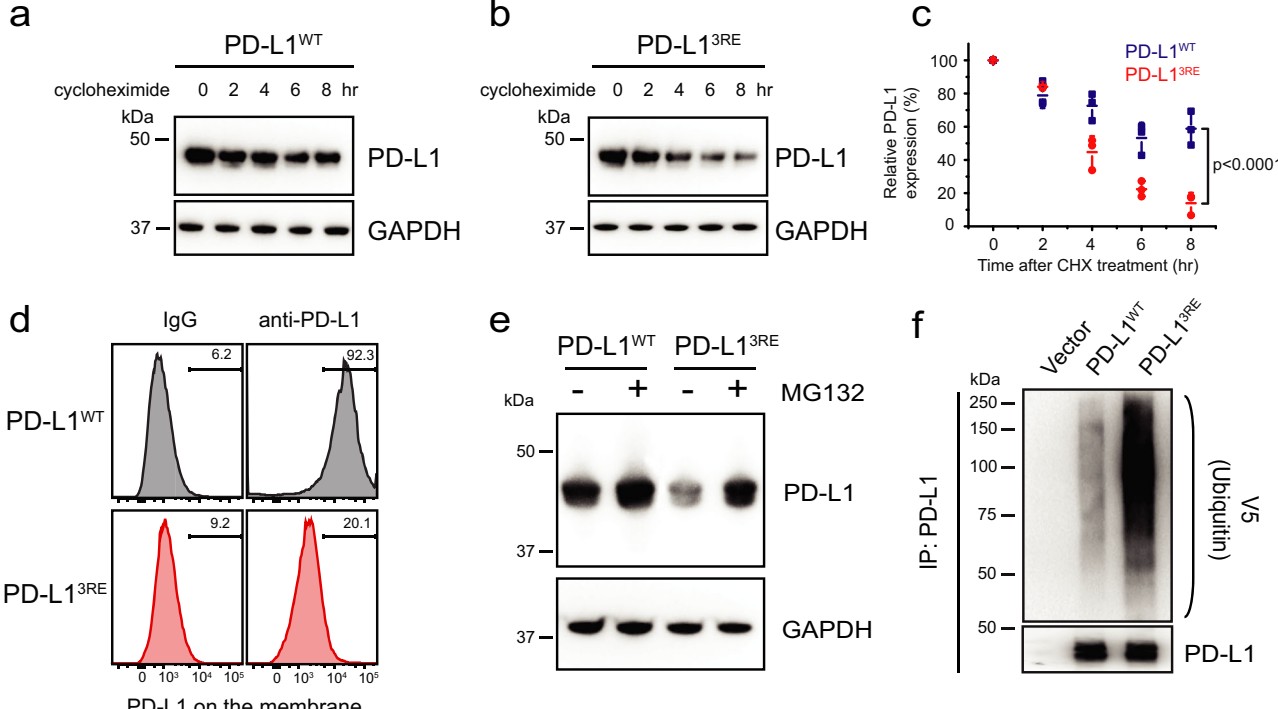

**Fig. 4 The 3RE mutation reduces cellular PD-L1 protein levels. a–c** Cellular levels of PD-L1$^{WT}$ or PD-L1$^{3RE}$ mutant. HEK293T cells expressing exogenous PD-L1$^{WT}$ or PD-L1$^{3RE}$ mutant were treated with 20 μM cycloheximide (CHX) for 2, 4, 6, or 8 h. The PD-L1 level was analyzed by western blot (**a**, **b**). The intensities of the PD-L1 protein bands on the blots were quantified by ImageJ analysis, and the statistical data shown in (**c**) are the mean ± SD ($n = 3$ independent experiments). $P$ values by two-sided Student's t-test were indicated. **d** Surface level of PD-L1$^{WT}$ or PD-L1$^{3RE}$ mutant in HEK293T cells determined by flow-cytometric analysis. **e** Cellular level of PD-L1$^{WT}$ or PD-L1$^{3RE}$ mutant in HEK293T cells treated with 10 μM of the proteasome inhibitor MG132 for 4 h (+ MG132) or not (- MG132) determined by western blot. **f** Ubiquitination of PD-L1$^{WT}$ or PD-L1$^{3RE}$ mutant in HEK293T cells examined by V5 immunoblotting after IP with PD-L1 antibody, along with the control cells that did not express these proteins (vector). All data shown above are representative of three independent experiments. Source data are provided as a Source Data file.

more degradation by the proteasome dependent on the ubiquitination. We further performed the CHX-chase assay and the ubiquitination examination on PD-L1$^{WT}$ and PD-L1$^{3RE}$ in RKO cells, showing consistent results that PD-L1$^{3RE}$ degraded faster than PD-L1$^{WT}$ with higher ubiquitination levels (Supplementary Fig. 6). Collectively, these observations suggest that the impaired membrane-binding capability of PD-L1$^{3RE}$ mutant promotes its ubiquitination-dependent degradation. Thus, the membrane association of PD-L1-CD enhances PD-L1 protein stability.

**Metformin promotes PD-L1 degradation by PD-L1-CD-membrane dissociation.** The critical role of PD-L1-CD in PD-L1 stability and degradation prompted us to search for therapeutics that can alter the membrane association of PD-L1-CD and hence control the PD-L1 level. Metformin has been reported to possess antitumor effects against various cancer types in recent years[31–33], though it has been a popular drug for the treatment of type 2 diabetes for long time[34]. However, how metformin acts on cancer cells remains controversial due to the use of supra-pharmacological concentrations in almost all previous studies[35]. Considering both arginine and metformin contain the same positively charged guanidinium group (Supplementary Fig. 7a), it is possible that metformin may compete with the arginine residues in PD-L1-CD for the lipid headgroups and thereby disrupt the membrane binding of PD-L1-CD, which allows the access to the enzymes that can ubiquitinate and degrade PD-L1.

To test this hypothesis, we first titrated metformin to CD$_{260-290}$ in the presence of lipid bicelles and monitored the changes of protein–membrane interaction using NMR spectroscopy. As the concentration of metformin increases, the resonances of CD$_{260-290}$ move toward the bicelle-free form in both $^{15}$N and $^{1}$H dimensions, indicating that the conformation of CD$_{260-290}$ shifts toward a non-membrane-bound state (Fig. 5a, b). We further performed PRE experiments to verify the role of metformin on the membrane association of CD$_{260-290}$. Titration of metformin to CD$_{260-290}$ in the presence of the DMPG/DH$^{6}$PC bicelles containing the lipophilic paramagnetic probe 16-DSA increased the recovery of resonance intensities of the membrane-bound residues (Fig. 5c), suggesting that metformin moves these residues away from the membrane and thus shielded this region from the paramagnetic probe. The liposome binding assay confirmed the negative effect of metformin on the membrane association of PD-L1-CD since increasing the concentration of metformin reduced the binding of PD-L1-CD to the liposomes (Supplementary Fig. 7b). Moreover, the {$^{1}$H}-$^{15}$N HetNOE values of the N-terminal 260-275 residues of CD$_{260-290}$ from the samples with bicelles are more positive and less dynamic than those from the samples without bicelles, while in the presence of metformin, the N-terminal half of CD$_{260-290}$ in bicelles gave low steady-state {$^{1}$H}-$^{15}$N HetNOE values as the samples in solution, indicating CD$_{260-290}$ is released from the membrane by metformin (Supplementary Fig. 7c).

Next, we tested the effects of metformin on PD-L1 in cells. In the absence and presence of metformin, the messenger RNA level for endogenous PD-L1 in RKO cells was similar (Supplementary Fig. 7d) but the protein level was reduced in the presence of metformin (Supplementary Fig. 7e). Moreover, the ubiquitinated protein level was higher when metformin was present (Supplementary Fig. 7f), which are consistent with previous studies[31,36].

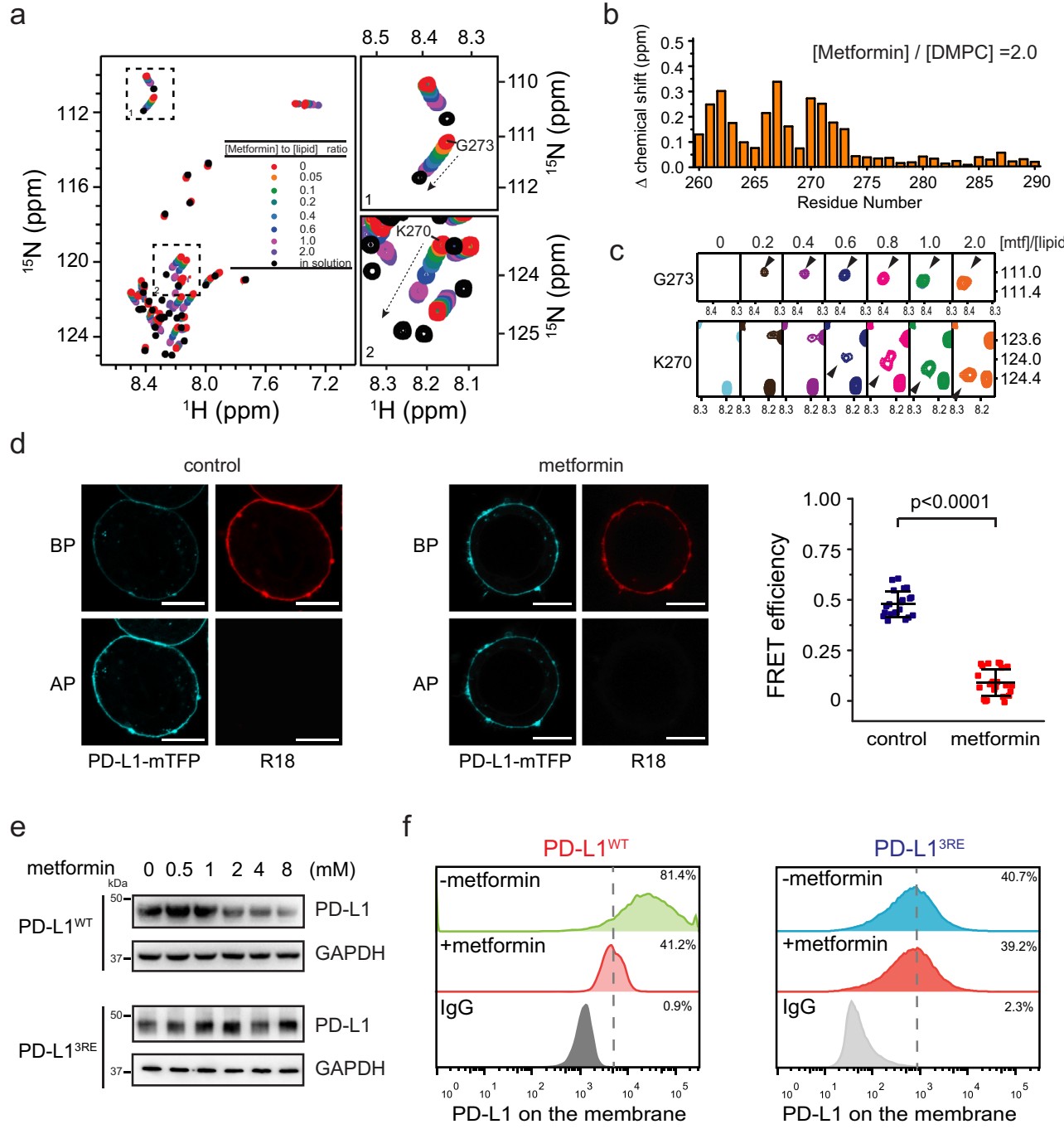

Importantly, the FRET imaging showed a significantly lower FRET efficiency after the metformin treatment of the RKO tumor cells expressing PD-L1$^{WT}$-mTFP (Fig. 5d). This direct observation of the membrane-binding status of PD-L1 via the C-terminal fluorescent tag indicates that metformin can induce the dissociation of PD-L1-CD from the plasma membrane.

Finally, we treated HEK293T cells expressing PD-L1$^{WT}$ or PD-L1$^{3RE}$ mutant with metformin and examined the PD-L1 protein level by western blot (Fig. 5e) and flow cytometry (Fig. 5f). The PD-L1$^{3RE}$ mutant showed similar protein levels with or without the metformin treatment, indicating that PD-L1$^{3RE}$ mutant that is less associated with the plasma membrane is less affected by metformin than PD-L1$^{WT}$ that is more associated with the membrane. The effects of a structural analog of metformin, moroxydine (Supplementary Fig. 8a), on the membrane

association of the cytoplasmic domain in vitro and the PD-L1 protein level in cells were also examined by NMR spectroscopy and western blot, respectively. The same results were observed with this metformin analog as the moroxydine titration shifted the resonances of PD-L1-CD in lipid bicelles toward that of the protein in solution (Supplementary Fig. 8b). And, the level of PD-L1 in RKO cells was reduced by the addition of moroxydine, though a higher amount of moroxydine than metformin was required to achieve the same effect (Supplementary Fig. 8a compared to Fig. 5e).

## Discussion

Recent studies suggest that intracellular expression of PD-L1 and the redistribution to the cell membrane may cause resistance to

**Fig. 5 Metformin prevents PD-L1-CD-membrane association to induce PD-L1 degradation. a** Metformin reduces $CD_{260-290}$ interaction with DMPG/DH$^6$PC bicelles. Superimposed 2D $^1$H-$^{15}$N TROSY-HSQC spectra of $CD_{260-290}$ in the bicelles (red) titrated with metformin at molar ratios of [metformin]:[DMPG] = 0.05 (orange), 0.1 (green), 0.2 (light blue), 0.4 (blue), 0.6 (dark blue), 1.0 (magenta), and 2.0 (purple). The addition of metformin shifts the correlations toward that of $CD_{260-290}$ in solution (black). The right panels 1–2 show the same spectral regions labeled on the full spectrum, highlighting the chemical shift changes for G273 and K270, respectively. **b** The comparison of chemical shift changes of $CD_{260-290}$ in DMPG/DH$^6$PC bicelles between molar ratios of [metformin]:[DMPG] = 0 and 2.0. **c** Metformin decreases $CD_{260-290}$ insertion into DMPG/DH$^6$PC bicelles. PRE of $CD_{260-290}$ spectral signals induced by the paramagnetic probe 16-DSA (2.5 mM) localized in the bicelle membranes were measured when the sample was titrated with metformin. The intensity recovery of broadened peaks for G273 (top) and K270 (bottom) at molar ratios of [metformin]:[DMPG] = 0, 0.2, 0.4, 0.6, 0.8, 1.0, and 2.0 are shown. **d** FRET-detected effect of metformin on PD-L1-CD-membrane interaction. RKO cells expressing PD-L1-mTFP were treated with 5 mM metformin for 2 h. Fluorescent images of representative cells from the vehicle (DMSO) control (left) and the metformin treated (middle) samples are shown. The statistical results of FRET efficiency are represented as the mean ± SD for $n = 24$ or 26 cells from the control and treated samples, respectively (right). $P$ values by unpaired two-sided Student's t-test are indicated. Scale bars, 5 μm. **e** Cellular level of PD-L1$^{WT}$ or PD-L1$^{3RE}$ mutant. HEK293T cells expressing PD-L1$^{WT}$ (top) or PD-L1$^{3RE}$ mutant (bottom) were treated with the indicated concentrations of metformin for 24 h and analyzed by western blot. Similar results were obtained from two other independent experiments. **f** Surface level of PD-L1$^{WT}$ or PD-L1$^{3RE}$ mutant in HEK293T cells determined by flow-cytometric analysis. After metformin (5 mM) treatment for 24 h, the HEK293T cells expressing PD-L1$^{WT}$ (left) or PD-L1$^{3RE}$ mutant (right) were analyzed by flow-cytometry using anti-PD-L1 antibody with IgG, immunoglobulin G, as the control. Source data are provided as a Source Data file.

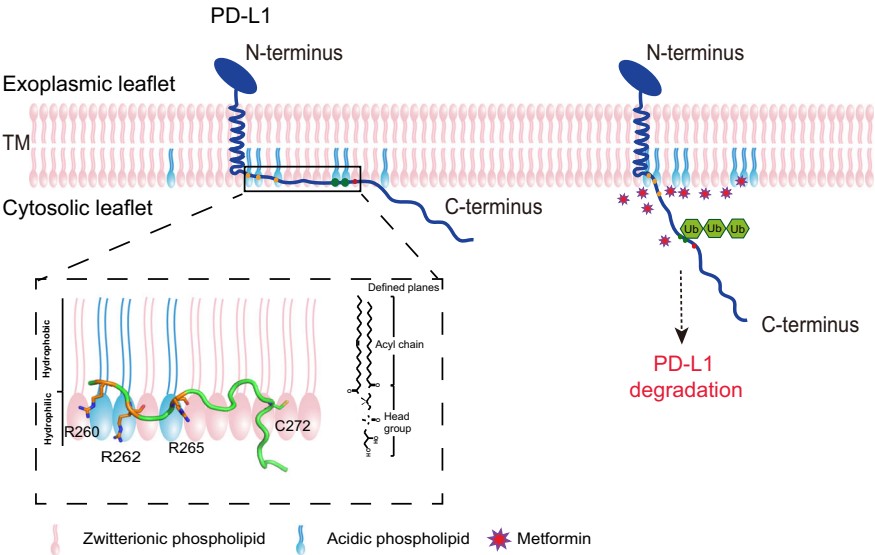

**Fig. 6 A model for regulation of PD-L1 degradation by the membrane association of its cytoplasmic domain.** PD-L1 on the cell surface is stabilized by the interaction of the cytoplasmic domain with the plasma membrane. Disruption of the electronic interactions between the acidic phospholipids in the membrane and the polybasic residues in the cytoplasmic domain unleashes the cytoplasmic domain from the membrane so that can be ubiquitinated, which marks the entire protein for the downstream degradation. The magnified region shows a ribbon diagram of $CD_{260-290}$ shallowly embedded in one leaflet of a lipid bilayer. The side chains of the three arginine residues and the cysteine at position 272 for palmitoylation are shown.

anticancer drugs and protect cancer cells from T cell-mediated immune surveillance[12,20,37]. Thus, alternative anticancer strategies are needed, and a recent focus is on the factors that control the cellular abundance of PD-L1[15,16]. The involvement of the cytoplasmic domain in multiple pathways that regulate cellular level of PD-L1 makes it a potentially target for exploring alternative cancer therapeutics in combination with the current immunotherapy. A step forward is to understand how the cytoplasmic domain of PD-L1 interacts with the plasma membrane and how this interaction regulates the PD-L1 level.

In this study, we characterized PD-L1-CD in a model membrane system using NMR spectroscopy. Previous studies showed that some disordered proteins display certain structure features when bound to membranes. For example, α-synuclein is predominantly a random coil in aqueous solution but adopts a helical secondary structure when interacts with negatively charged membranes[38]. In our case, no obvious secondary structure forms upon the binding to lipid bicelles, which is similar to the observations on CD28 and CD3ε cytoplasmic tails[24,26]. We

further demonstrated that the electrostatic interactions between a positively charged patch in the cytoplasmic domain and negatively charged phospholipids are critical to the membrane interaction and to the degradation of PD-L1 protein in tumor cells (Fig. 6). In the presence of acidic lipids, the three arginine residues adjacent to the TM domain project their side chains into the membrane (Fig. 6, magnified insert). This orientation facilitates the ionic interactions between the positive guanidinium groups of the arginines and the negative phosphate moieties in the lipid headgroups, when substituting the arginines by glutamates completely abolishes the membrane association (Fig. 3a). In contrast, replacement of the arginines with lysines does not reduce the membrane binding (Supplementary Fig. 5e, f), indicating that the electrostatic attraction is the dominant binding force. K263 within this basic patch may also contribute to the membrane association of PD-L1-CD, though we did not investigate its role due to its relatively weak NOE signals to lipids. In addition, the position of the polybasic residues in PD-L1-CD matters since mutations of K270 and K271 that are farther away

from the TM domain do not disrupt the membrane association of PD-L1-CD (Supplementary Fig. 5c, d). Together, only the cluster of basic residues in PD-L1-CD located in the closest proximity to the TM domain is essential for the membrane association.

The disruption of PD-L1-CD association with the plasma membrane by the triple arginine mutation leads to a reduction of PD-L1 cellular level. However, it is not clear which of the multiple pathways that regulate PD-L1 level is affected by the PD-L1-CD-membrane dissociation. One possibility is that the membrane partition of the polybasic residues may cause steric hindrance and hence limit the accessibility of themselves or neighboring residues to other downstream proteins, such as SPOP or HIP1R, which promote the ubiquitination-dependent proteosomal or lysosomal PD-L1 degradation in the cells. On the other hand, palmitoylation of PD-L1 has been reported to modulate the protein stability by blocking its degradation[16]. PD-L1 is palmitoylated by DHHC3 acetyltransferase at C272[16], seven residues away from the basic membrane-binding patch. Previous crystal structures of the DHHC enzymes showed that the substrate palmitoyl-CoA inserts into the cavity formed by the enzymes within the membrane[39]. If C272 were located within the membrane as our lipid NOE signal indicated (Fig. 2b), it would be accessible to the palmitoyl-CoA in the DHHC3 enzyme. When PD-L1-CD is dissociated from the membrane, C272 is less accessible to the enzyme and substrate, which can significantly decrease the palmitoylation and hence the stability of PD-L1. Another possibility is that the effects of PD-L1-CD-membrane dissociation on PD-L1 degradation are a combinational outcome from multiple pathways including promoted PD-L1 degradation by increasing SPOP or HIP1R recruitment to PD-L1-CD and reducing DHHC3 palmitoylation. Further investigations are needed to examine these possibilities.

Calcium is important for cellular signaling that exerts regulatory effects on many enzymes and proteins. Previous studies showed that a calcium influx in T cells can disrupt the membrane interactions of the CD3ε[25,40] and CD28[24] tails by shielding the acidic lipid head groups, which releases the motifs in the cytoplasmic domains and mediates the downstream signaling. A high dietary calcium intake has been found to reduce the risk of one or more types of cancer, such as colorectal cancer and rectal cancer[41,42], whereas another study suggested that a high calcium intake may actually increase the risk of prostate cancer[43]. Although the relationships between calcium intake and cancer risk have not been consistent and the exact mechanism about how calcium may help reduce the risk of some cancer types is unclear, we guess one of the possible mechanisms for the reduced risk by the high calcium intake is that the positive charges of calcium can neutralize the membrane charges as well as what was observed in T cell signaling[24,44] and release PD-L1-CD from the membrane, which leads to the reduction of PD-L1 levels in some cancer cells.

The FDA-approved type 2 diabetes drug metformin has been reported to exert substantial antitumor effects in several clinical studies[45,46]. However, its mechanism of action is only partially understood. Zhang et al reported that metformin activates Hippo signaling pathway to regulate PD-L1 expression[36]. Cha and colleagues recently revealed that metformin can activate the AMPK-dependent phosphorylation of PD-L1 at S195 and thus induce the abnormal PD-L1 glycosylation and degradation through an Endoplasmic-reticulum-associated protein degradation (ERAD) pathway[47]. Strikingly, our study uncovered a previously unknown effect of metformin on PD-L1, shedding light on a more complex mechanism. We observed that metformin is able to disrupt the interaction between the polybasic juxtamembrane region of PD-L1-CD and the acidic cytoplasmic surface of the plasma membrane using NMR technology and FRET measurements. Consistent with the notion that metformin disrupts the PD-L1-CD-

membrane interaction, we detected a minor FRET reduction from the PD-L1$^{3RE}$ mutant to the membrane in the presence of metformin since its cytoplasmic domain was already released from the plasma membrane by the mutation. Moroxydine, a chemical analog of metformin, can also cause the dissociation of the cytoplasmic domain from the membrane. The PD-L1-CD-membrane disruption caused by these two compounds further reduced the PD-L1 abundance significantly. A common mechanism for the metformin-kind of molecules to inhibit the PD-L1 level is by electrostatically disrupting the membrane association of PD-L1-CD that is critical to the PD-L1 cellular levels. The above results have led to a unique strategy that the disruption of PD-L1-CD-membrane association can be explored for the cancer therapeutics. However, metformin treatment on cancer cells usually requires high doses. Many previous studies showed that only millimolar concentrations of metformin led to a significant reduction of the cellular abundance of PD-L1[31,47]. And a number of clinical studies also used metformin in high doses of 1,500–2,250 mg per day to reduce the risk of cancer[48]. The concentration of metformin used in our study is also high (mM range) due to the nonspecific effects of metformin. Therefore, more specific molecules targeting on PD-L1-CD-membrane interactions should be designed and screened to increase the efficacy in the treatment of cancer.

While many open questions remain regarding how PD-L1-CD-membrane interactions regulate other components and downstream signaling pathways, our study suggests a physiological role for acidic phospholipids in regulating the PD-L1 level in tumor cells. Acidic phospholipids act not only as a membrane scaffold, but also involve in the molecular regulation of tumor cell survival. The NMR studies of the membrane-bound PD-L1-CD showed how basic residues in the cytoplasmic tail interact with a membrane to regulate a receptor protein conformation in the membrane, which in turn regulates the receptor stability and hence its function in a signal transduction. We note that some cancer derived mutations found in this cytoplasmic region might enhance the membrane association to escape the immunosurveillance. For instance, D276Y and T290M can increase the residue hydrophobicity with the replacements and stabilize PD-L1 with stronger hydrophobic membrane interactions. R260C might provide a new position for the palmitoylation and therefore improve the stability. It's not clear whether R262K enhances the membrane interactions of PD-L1, since arginine has a guanidino group but a shorter acyl chain. Further studies of these mutations in biochemical and cell biological systems are required to confirm or refute the expectations. Interestingly, a broad range of transmembrane receptors also have short cytoplasmic domains that are highly basic. For example, the isoelectric point (pI) for B7 family ligands CD80, CD86 and PD-L2 is 11.47, 9.72, and 10.30, respectively. Therefore, we expect that the electrostatic interactions between the receptors and lipids regulating the protein stability and degradation may occur in a general manner, and our study brings valuable insights into the lipid regulation of these transmembrane receptors.

## Methods

**Reagents and cells**. Lipids and detergents (DMPC, DMPG, DMPE, DMPS, POPG, DH$^6$PC, Sph and PS (Brain, Porcine)) were from Avanti Polar Lipids. Stable isotopes for NMR spectroscopy experiments were from Cambridge Isotope Laboratories. Anti-human PD-L1 antibodies (ab213524), Alexa488-conjugated anti-IgG (ab150077) and anti IgG-HRP (ab97051) were from Abcam. Anti-V5 antibodies (96025) were from Thermo and FITC anti PD-L1 antibodies (393606) were from Biolegend. MG132 (HY-13259) and metformin (HY-B0627) were from MedChem Express (MCE). The HEK293T cell line was a generous gift from Liming Sun (SIBCB). The RKO cell lines for expressing PD-L1 were generous gifts from Jie Xu (Fudan University), other cell lines used in the paper were from Cell Bank of Chinese Academy of Sciences.

**Expression and purification of CD$_{260-290}$ and its mutants**. The cytoplasmic domain of human PD-L1 (residues 260–290), named as CD$_{260-290}$, was fused to an N-terminal 8 × His tag, a Small Ubiquitin-like Modifier (SUMO) protein and a PreScission protease (3 C) in the pET28a vector. The construct was transformed into *Escherichia coli* BL21 (DE3) cells and grown at 37 °C in M9 minimal media until the culture reached an optical density at 600 nm (OD$_{600}$) of 0.6–0.8. One or more stable isotopes were supplemented to the growth media according to the NMR experimental requirements. Cells were cooled to 25 °C before the induction with 0.5 mM isopropyl β-D-thiogalatopyranoside (IPTG) at 25 °C overnight. Full deuteration of CD$_{260-290}$ was achieved with the growth in 99.9% D$_2$O (Sigma Aldrich) with deuterated glucose (Cambridge Isotope Laboratories).

The expressed fusion protein was extracted and purified by nickel affinity resins (Thermo Fisher) followed by 3 C cleavage at 4 °C for 14 h to remove the His and SUMO tag. The CD$_{260-290}$ protein was further purified by reverse-phase HPLC with Zorbax 300SB-C18 PrepHT column (Agilent) using an elution gradient from 10% (v/v) acetonitrile with 0.1% (v/v) trifluoroacetic acid (TFA) to 80% (v/v) acetonitrile and 0.1% (v/v) TFA. The fractions corresponding to pure CD$_{260-290}$ peptide were identified by MALDI-TOF mass spectrometry and SDS–PAGE analysis. All the mutants were expressed and purified following the same protocols.

**Reconstitution of CD$_{260-290}$ into bicelles**. The NMR samples without bicelles were prepared by directly dissolving CD$_{260-290}$ and variants (0.8–1 mg) in 25 mM MES pH 6.5, 10% D$_2$O. To reconstitute CD$_{260-290}$ and variants in bicelles, 0.8–1 mg of purified and lyophilized proteins were mixed with 10 mg POPG, DMPG (protonated or deuterated from Avanti Ploar Lipids) or DMPC and dissolved in hexafluoroisopropanol. The mixture was slowly dried to a thin film under nitrogen stream, followed by overnight lyophilization. The dried thin film was redissolved in 0.5 mL of 25 mM MES pH 6.5 buffer containing 37 mM DH$^6$PC (protonated or deuterated from Avanti Polar Lipids). The POPG:DH$^6$PC, DMPG:DH$^6$PC or DMPC:DH$^6$PC ratio was measured by 1D NMR to verify the q value. The final NMR sample contained 0.4–0.5 mM CD$_{260-290}$ or variants, ~ 30 mM POPG, DMPG or DMPC, ~ 37.5 mM DH$^6$PC, 25 mM MES pH 6.5 and 10% D$_2$O.

**Assignment of NMR resonances in bicelles**. All NMR spectra were acquired at 30 °C by Topspin on Bruker Avance III 600 MHz or 900 MHz spectrometers equipped with cryogenic probes and by VnmrJ Biopack on Agilent DD2 800 or 700 MHz spectrometer equipped with triple-resonance cold probes. Detailed parameters for the NMR experiments are given in the Supplementary Table 1. NMR data were processed using NMRPipe[49] and analyzed by Sparky[50] and XEASY[51]. Sequence specific assignment of backbone chemical shifts was accomplished by performing a suite of standard triple resonance experiments, including the TROSY version of HNCA, HN(CO)CA, HN(CA)CO, HNCO and HNCACB on a ($^{15}$N, $^{13}$C, 85% $^2$H)-labeled sample at $^1$H frequency of 600 MHz. These spectra employed a non-uniform sampling scheme in the indirect dimensions and were reconstructed using Sparse Multidimensional Iterative Lineshape-Enhanced (SMILE) algorithm interfaced with NMRPipe. Aliphatic side chain assignments relied on (H)CCH-TOCSY and H(C)CH-TOCSY spectra to measure the homo- and heteronuclear J couplings from quantitative J correlation[52,53] on a $^{15}$N/$^{13}$C-labeled sample. 98% of the C and H resonances for all backbone and side chains have been assigned.

The backbone dynamics experiments were performed using an Agilent DD2 800 MHz spectrometer on the $^{15}$N/$^{13}$C-labeled sample[54], in which the NOE spectrum was acquired with a 7 s recycle delay followed by a 5 s saturation and the reference spectrum was collected with no saturation and a 12 s recycle delay. The intensities of the NOE peaks were estimated by the Sparky program[50], and the NOE ratios of the two states were calculated using the peak intensities in the presence and absence of proton saturation.

**NMR characterization of membrane-bound CD$_{260-290}$**. Protein intramolecular distance restraints for membrane-bound CD$_{260-290}$ were derived from cross-peaks with a simultaneous $^{15}$N and $^{13}$C-NOESY-HSQC ($\tau_{NOE}$ = 120 ms) experiments[55,56] on a $^{15}$N/$^{13}$C-labeled sample. Peak analysis of the NOESY spectra was generated by interactive peak picking with the program Sparky[50].

To position the dynamic PD-L1-CD in the lipid bilayer, the protein–lipid distance restraints were obtained from a 3D $^{15}$N-edited NOESY-TROSY-HSQC spectrum ($\tau_{NOE}$ = 200 ms) recorded with a $^{15}$N, $^2$H-labeled protein sample in bicelles made of regular POPG and deuterated DH$^6$PC (q = 0.8), which allowed the measurement of exclusive protein–lipid NOEs between the peptide backbone amide protons and POPG aliphatic protons. The protein–lipid NOEs were then categorized into four groups according to the protein positions to the lipid. Three parallel planes were used to represent a leaflet of the POPG bilayer: the POPG head-group glycerol (y = −22 Å), the hydrophobic acyl chain (y = −11 Å) and the methyl tail (y = 0 Å).

**Lipophilic PRE and solvent PRE analysis of membrane-bound CD$_{260-290}$**. We cross-validated the NOE-derived membrane insertion using paramagnetic probe titration (PPT) method[28]. The solvent PRE measurements were performed using a 0.4 mM $^{15}$N-labeled CD$_{260-290}$ in DMPG bicelles with q = 0.8. The water-soluble and membrane inaccessible paramagnetic agent, Gd-DOTA (Macrocyclics, Inc.)

was titrated into the bicelle sample at different concentrations, including 0, 0.1, 0.2, 0.4, 0.6, 1, 1.5, 2, 3, 6, 10, 15, 20, and 30 mM. At each titration point, a 2D $^1$H-$^{15}$N TROSY-HSQC spectrum was recorded at a 700 MHz Agilent spectrometer. The recovery delay was set to 3 s. The residue-specific PRE is defined as the ratio of peak intensity in the presence (I) and absence (I$_0$) of the paramagnetic agent. Peak intensities were measured at peak local maxima by quadratic interpolation to identify peak centers. For individual peaks, Origin was used to fit I/I$_0$ vs. Gd-DOTA concentration to the exponential decay to derive the residue-specific PRE amplitude (PRE$_{amp}$) by the following equation, in which τ is the decay constant[28].

$$\frac{I}{I_0} = 1 - \text{PRE}_{amp}\left(1 - e^{-\frac{[Gd-DOTA]}{\tau}}\right) \qquad (1)$$

PRE$_{amp}$ is the indicator of PRE effects of the probe on the protein, as the higher PRE$_{amp}$ values indicate the protein is more close to the PRE probe. The lipophilic PRE measurements were performed using a 0.4 mM $^{15}$N-labeled CD$_{260-290}$ in DMPG bicelles with q = 0.8. A stock solution of lipophilic paramagnetic agent 16-DSA (Sigma-Aldrich) was prepared at 40 mM concentration in the same NMR sample buffer to prevent changes of q value in the bicelles upon addition of the titrant. The progress of the titration was monitored by measuring a set of 2D $^{15}$N-TROSY-HSQC spectra at each of the following 16-DSA concentrations: 0, 0.1, 0.25, 0.5, 1, 2.5, 3, and 5 mM. The residue-specific PREamp was determined by fitting the peak intensity decay as a function of [16-DSA].

**Liposome-binding assays**. Mixed phospholipids were dissolved in chloroform with indicated compositions (10 mg). The solvent was evaporated under a stream of nitrogen to achieve a thin film. Followed by an overnight lyophilization, 500 μL extrusion buffer (20 mM MES, pH 6.5) was added to the dried lipid mixture. Liposomes were generated by extrusion of the hydrated lipids through a 0.1 μm polycarbonate filter (610005, Avanti Polar Lipids Inc.) 50 times using the Mini-Extruder device (Avanti Polar Lipids Inc.). CD$_{260-290}^{WT}$ or mutants (50 μM) were incubated with the indicated liposomes (5 mM lipids) at room temperature for 30 min in a total volume of 200 μL, respectively. Samples were centrifuged at 4 °C for 1 h at 100, 000 g. The supernatant and pellet fractions were analyzed by SDS–PAGE followed by Coomassie blue staining.

**Confocal imaging analysis**. For immunofluorescence, RKO cells were seeded at approximately 50% confluence in a glass bottom dish (150680, Thermo). After removal of culture medium, the dish was washed twice with PBS. Then, cells were fixed with 4% Paraformaldehyde (158127, Sigma) for 20 min and washed three times with PBS. Cells were then permeabilized with 0.2% Triton X-100 in PBS and blocked in 5% BSA in PBS for 1 h at room temperature. FITC anti-PD-L1 antibodies were diluted in the blocking buffer (1:100) and incubated cells overnight at 4 °C. After rinsed by PBS three times, cells was treated by SlowFade Diamond Antifade reagent (S36968, Thermo) and sealed by a coverslip. Cells were observed with Zeiss LSM710 confocal microscope (Carl Zeiss) fitted with a 100× oil immersion objective. Micrographs were captured by means of confocal software (ZEN system 2012 Black Edition, Zeiss). All confocal images are representatives of three independent experiments.

CD$_{260-290}^{WT}$-GFP and mutants expressed in HEK293T cells were plated on coverslips in glass bottom dish in complete media. Cells were fixed with 4% paraformaldehyde for 20 min at room temperature. Cells were washed three times with PBS for each step. After treated by SlowFade Diamond Antifade reagent, cells were observed on Zeiss LSM710 confocal microscope fitted with a 100 × oil immersion objective.

**Western blot and ubiquitination assays**. For western blot analysis, cells were lysed in RIPA (50 mM Tris-HCl pH 7.4, 150 mM NaCl, 1% NP-40) buffer supplemented with protease inhibitors cocktail (11697498001, Merck) after PBS washing. Protein concentrations were measured by bicinchoninic acid (BCA) assay. Equal amounts of protein were resolved by SDS–PAGE and immunoblotted with antibodies (PD-L1 antibody, 1:5000; GAPDH antibody, 1:8000) overnight at 4 °C. The membrane was then washed three times with TBS-T (50 mM Tris, 1.37 mM NaCl, 2.7 mM KCl, pH 8.0; 0.1% Tween 20) buffer. The membrane was incubated with HRP-conjugated goat anti-rabbit IgG antibodies (1:10000) for 1 h at room temperature. After washed five times with TBS-T, the western blot bands were detected by using an ECL western blotting substrate (T7101A, Takara). The protein bands from western blot were quantified by ImageJ.

For ubiquitination assay, HEK293T cells were transfected with the indicated plasmids of PD-L1 or mutants and V5-ubiquitin via lentiviral transduction and stable clones were isolated by puromycin and single-cell sorting. Cells were treated with 10 μM MG132 for 6 h before harvesting. Cells were then lysed in RIPA lysis buffer and subjected to Co-IP with anti-PD-L1 antibodies and protein G-conjugated agarose beads (10003D, Thermo) followed by immunoblotting analysis with anti-V5 antibody (1:3000).

**Flow cytometry for detection of cell-surface PD-L1**. For the detection of cell-surface PD-L1, cells were harvested and suspended in 200 μL PBS buffer after washing twice and incubating with anti-human PD-L1 antibody (1:100) at 4 °C for 30 min. After washing twice by PBS, the cells were stained by Alexa488-conjugated

secondary antibodies (goat-anti-rabbit IgG) were diluted in PBS (1:500) at 4 °C for 30 min. After washing twice by PBS, stained cells were collected by BD FACSDiva software in flow cytometry (BD, LSRFortessa). Data were analyzed by FlowJo Software. The gating strategy is shown in Supplementary Fig. 9.

**Fluorescence resonance energy transfer measurements and analysis**. Human PD-L1 containing a C-terminal mTFP tag (a gift from Prof. Chenqi Xu's lab) with a GSS linker was constructed. The constructs were transfected into RKO cells via lentiviral transduction and stable clones were isolated by single-cell sorting. Cells were injected into a flow chamber set-up treated with poly-lysine. After staining cells with 1 μM octadecyl rhodamine B (R18, O246, Thermo) membrane-labeling dye for 5 min on ice, FRET efficiencies between the C-terminal mTFP (the donor) and the membrane dye R18 (the receptor) were measured by fluorescence microscopy (Zeiss LSM 710) using the dequenching method that mTFP was excited at 458-nm and visualized by detection at 470–550 nm; R18 was excited at 561-nm and visualized by detection at 570–650 nm. Fully photobleached R18 signal of whole cells by the 561-nm laser set at 100% efficiency for 60 s. The mTFP fluorescence signals were measured before and after R18 photobleaching three times and FRET efficiencies were calculated using the following formula:

$$\text{FRET efficiency} = (\text{FL}_{AP} - \text{FL}_{BP})/\text{FL}_{AP} \tag{2}$$

where $\text{FL}_{AP}$ and $\text{FL}_{BP}$ represent mTFP fluorescence after and before R18 photobleaching, respectively.

In the FRET detected lipid effects on PD-L1-CD-membrane interaction assay, $1 \times 10^6$ RKO/PD-L1-mTFP cells were seeded in a 6-well plate and cultured in RPMI-1640 medium treated with dimethylsulfoxide (DMSO, negative control), 5 μM phosphatidylserine (PS) or 2 μM sphingosine (Sph) for 2 h, respectively. Cells were collected and fixed into a flow chamber treated with poly-lysine. After staining cells with 1 μM R18, fluorescent images of the cells were taken to detect the mTFP or R18 dye emission before or after photobleaching (BP or AP) the R18 dye. FRET efficiencies were calculated by Eq. 2.

**Metformin and moroxydine titration assay**. The metformin titration experiments were used to detect metformin effects on $\text{CD}_{260-290}$ in DMPG/DH$^6$PC bicelles. 0.4 mM $^{15}$N-labeled $\text{CD}_{260-290}$ was reconstituted in q = 0.8 DMPG/DH$^6$PC bicelles. A stock solution of 2 M metformin was added to the NMR sample at the molar ratios of [metformin]:[DMPG] = 0, 0.05, 0.1, 0.2, 0.4, 0.6, 1.0, 2.0. A 2D $^1$H-$^{15}$N TROSY-HSQC spectrum was recorded at each concentration of metformin. Moroxydine was titrated to $\text{CD}_{260-290}$ following the same protocol. To further verify the effect of metformin on $\text{CD}_{260-290}$, PRE recovery experiment was performed using 0.4 mM $^{15}$N-labeled $\text{CD}_{260-290}$ reconstituted in DMPG/DH$^6$PC bicelles with 2.5 mM 16-DSA. Metformin was then titrated using the same method mentioned above at molar ratios of [metformin]:[DMPG] = 0, 0.2, 0.4, 0.6, 0.8, 1.0, 2.0.

**Reporting summary**. Further information on research design is available in the Nature Research Reporting Summary linked to this article.

## Data availability
$^1$H, $^{13}$C, and $^{15}$N chemical shifts have been deposited in the Biological Magnetic Resonance Bank under accession number BMRB 36293 (Membrane-bound human PD-L1 cytoplasmic domain), related to the PDB code 6L8R. Source data are provided with this paper.

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

## Acknowledgements

We thank the staffs from Large-scale Protein Preparation System/Nuclear Magnetic Resonance System/Mass Spectrometry System/Integrated Laser Microscopy System/ Molecular Imaging System at National Facility for Protein Science in Shanghai, Zhangjiang Laboratory (NFPS, ZJLab), China for providing technical support and assistance in data collection and analysis. This work was supported by grants from National Natural Science Foundation of China (31861133009 and U1732125), National Key R&D Program of China (2017YFA0504804), Key Research Program of Frontier Sciences, CAS (QYZDB-SSW-SMC043), and CAS Major Science and Technology Infrastructure Open Research Projects to B.O. and Institutional Development Award from the National Institute of General Medical Sciences of US National Institutes of Health (P20GM103640) to J.L.

## Author contributions

B.O. and M.W. conceived of the study; Y. C. and Q. W. prepared NMR samples; M.W. and Y.C. performed NMR experiments and data collection with the help of H.X.; M.W. and B.W. analyzed the NMR data; M.W., Y.C., T.X., and R.C. performed the biochemical and imaging experiments with the help of X.L. and Y.Y.; J.L., J.X. and C.X. provided extensive discussions; B.O. and M.W. wrote the paper, and all authors, especially J.L., contributed to editing of the manuscript.

## Competing interests

The authors declare no competing interests.
