## [Peer Review File · Nature Communications]

REVIEWER COMMENTS

Reviewer #1 (Remarks to the Author):

This paper by Wen et al. describes an important scientific question on the stability of PD-L1, through the interaction of its cytoplasmic tail with the membrane. Though the results could be potentially interesting, the paper as such has some fundamental problems. The one at the top of the list is the structure.

The authors have solved the of the isolated cytoplasmic tail in POPG and report a structure with an RMSD of 0.5 Å. What is the meaning of this structure? Are the authors trying to say that this is the structure of the cytoplasmic tail in a biological context? The structure looks very odd, it has a trapezium-shaped backbone at places. I am not sure if this will fit to any of Ramachandran parameters. Just because we can get obtain a tight structure to fit NOEs it does not mean it is correct.

Critical information as to how many long-range NOEs (i to i+4 and above) is missing.

I would request the authors to look at figure 2D and figure 2E, which are right below one another. Figure 2D shows a tight bundle of structures and figure 2E the hetero-NOE data shows that the section is for which they reported a tight bundle of structures (2D) is indeed highly dynamic. How do they reconcile these opposing results? The backbone structure in figure 2F right after R265 looks very unusual. What drives such an unusual structure? This square/trapezium-shaped backbone is seen again in the C-terminal. This is a serious problem that needs to be addressed.

I would suggest removing the structure and keeping the NMR data. The authors have to update their structure-calculation table with the number of long-range and short-range NOEs. They claim they have used dihedral restraints from TALOS. TALOS calculates dihedral restraints using chemical shift data that have been correlated to secondary structures and here there is no secondary structure. How can TALOS constraints be used?

It would have been prudent to have solved the structure with the transmembrane helix, which will position the cytoplasmic tail in the membrane. Without that this structure holds no meaning as presented.

Other Concerns.

a) Fig1. The authors should provide a plot of chemical shift perturbations as well as peak intensities plotted against the protein sequence. If the claim that the cytoplasmic tail is getting embedded in the membrane (DMPG) with a well-defined structure is correct, then they should observe a remarkable reduction in intensities.

b) The spectrum of the cytoplasmic tail in solution seems to have remarkable dispersion compared to the DMPG spectrum, especially look at the peak at 1H-7.7ppm and 15N-121 ppm. It is missing from the DMPG spectrum. What is the explanation for this? Why does it reappear in the POPG spectrum?

c) One problem with the experiment is the DMPG/DHPC mix used for bicelles. All the acidic lipids will cluster on the low curvature sides of the bicelles likely providing an effective 100% acidic surface, far from plasma membrane acidity. It would be better to do the experiment with 25%DMPG-75%DMPC/DHPC bicelles. Or even better, a POPG/POPC mix. This is absolutely necessary to validate that the structure determination is close to that inside cells and not an artifact of the artificially high acidic lipid content.

d) Another potential problem with that experiment is the extremely low ionic strength of the buffer (25 mM MES, no salt) which is also far from cell conditions. A control experiment with 100-150 mM NaCl should be performed to see if the electrostatic interaction still takes place.

e) Was there a reason to change from DMPG to POPG for structure determination?

f) It was not very clear to me what secondary structure may or may not form upon lipid binding. A lot of similar membrane interacting protein form imperfect alpha-helices but it is not the case here? The authors could compare with the structure of alpha-synuclein bound to lipids (Ulmer et al 2005 JBC 280).

g) The authors should discuss the effect of calcium ions if anything has been reported. Influx of calcium is likely to disrupt membrane interactions by shielding acidic lipids as well (Zhang et al 2014 BBA 1838).

h) The authors state "A stock solution of 2 M metformin was added to the NMR sample at

the molar ratios of [metformin]:[DMPG]= 0, 0.05, 0.1, 0.2, 0.4, 0.6, 1.0, 2.0. “ Is this not a very high metformin concentration? Is this even biologically relevant? If we want to connect to the biology that they are implying in the discussion section, we should limit our discussions of results to metformin concentrations of ~ 40uM.

NMR experiments with arginine will be a welcome addition to the manuscript.

i) The authors need to show an heteroNOE experiment in the non-bicell (solution) state to show the difference due to membrane association.

j) In the introduction the authors write “ Besides the multiple regulatory roles of the cytoplasmic domain, several cancer derived mutations were found within this domain, including R260C, R262K, D276Y, and T290M, alluding its critical role in cancer cell survival”

In the light of their studies what is effect of these mutations on membrane association and the stability of the PD-L1.

Minor points.

1) Please consider working on the title. PD-L1 is repeated in a span of two words.

2) Equation 1 needs a definition of all the terms. What is tau? For those of us who are not familiar with this way of representing the data, the authors have to explain what PREamp means, especially in this case the higher number means PRE effect.

3) Figure 5, the CSP has to be plotted as a graph for each residue so that we can clearly see what is happening.

4) What is the saturation time in the heteroNOE experiment?

5) The authors state “Protein intramolecular distance restraints for membrane bound PD L1 CD structure calculations were derived from cross peaks in a simultaneous ^{15}N and ^1C NOESY HSQC (τ NOE= 120)”. What is a simultaneous NOESY? Are they referring to time-shared NOESY? Is the reference (29) the correct one here ?

6) The description of the NMR method is very minimal and should be expanded to contain details of the 3D experiments, how many points were collected, what mixing times were used for the TOCSY experiment, how was the data processed etc.

7) They used a deuterated sample for backbone experiment, while a protonated sample for the side-chain experiment, which is more demanding in terms of relaxation. Considering the protein is very flexible, was the deuteration actually needed in the first place.

Reviewer #2 (Remarks to the Author):

Electrostatic membrane association of the cytoplasmic domain of PD-L1 regulates PD-L1 degradation.

Maorong Wen et al.

Reviewer Summary

The authors employ a combination of NMR techniques, in vitro assays and FRET confocal imaging to investigate the regulation of PD-L1 stability via interaction of the PD-L1 cytoplasmic tail domain (CD) with plasma membrane lipids. Their structural analysis of the PD-L1 cytoplasmic tail (residues 260 - 290) bound to lipid bicelles identified three specific arginine residues (R260, R262 and R265) that associate directly with acidic lipids in the membrane. Using CD mutants of PD-L1 they demonstrate that loss of membrane association results in less stable PD-L1 that has higher levels of ubiquitination and is consequently degraded faster inside cells. Additionally, they show that the FDA-approved drug Metformin, which contains a positively charged guanidinium group, can block membrane association of WT PD-L1-CD and subsequently decrease PD-L1 protein levels (total and cell-surface expressed). Together their data suggest a novel mechanism by which membrane association of the cytoplasmic domain of PD-L1 regulates PD-L1 turnover, thereby controlling cell-surface levels of PD-L1. Overall the experiments are well designed, well presented and the results are internally consistent with each other. However, the manuscript would benefit from additional consideration of how electrostatic association between the CD and membrane lipids relates to previously identified mechanisms that regulate PD-L1 turnover (SPOP dependent proteasome degradation, Cu²⁺ influx etc.) and also what role (if any) PD-1 or B7-1 engagement may play in this process. Detailed comments follow below:

Reviewer Comments and Suggestions

1) This manuscript does not consider what role receptor engagement might play in regulating the cytoplasmic domain even though PD-1 engagement drives the immunosuppressive capacity of tumor

cells. Residue R260 in PD-L1 is located right at the C-terminal end of the transmembrane helix; in fact prediction algorithms place this residue within the transmembrane domain (239 – 261). This suggests the possibility that PD-1 engagement might trigger repositioning of the cytoplasmic tail domain via a mechanically transduced conformational change translated through the transmembrane domain. The authors could directly address potential PD-1 dependent effects using their in vitro FRET and HEK cell expression experiments, both of which utilized expression of full-length PD-L1 constructs.

2) PD-L1 also interacts with CD80 in cis and formation of the cis complex has been shown to inhibit PD-1 activity in vivo. Although CD80 is most commonly down regulated in solid tumors, both on tumor cells and on tumor associated macrophages, it is possible that in the context of normal APC cells, B7-1 binding in cis affects overall PD-L1 stability. Both CD80 and CD86 also have short cytoplasmic domains that are highly basic (pI of 11.47 and 9.72 respectively). Do the authors suggest that this is a general mechanism for all of the B7 family ligands? Does the overall charge of the cytoplasmic domain correlate with stability?

3) The cytoplasmic domain of PD-L2 is even more basic than that of PD-L1 (pI of 10.30 compared to 9.52 for PD-L1). The authors' hypothesis would suggest that PD-L2 should be more membrane associated and therefore more stable than PD-L1. These features might provide another mechanism for driving specific PD-L1 and PD-L2 functionality.

4) There are a number of other mechanisms that have been shown to play a role in regulating PD-L1 stability and turnover (i.e. SPOP binding, CSN5 binding, HIP1R binding, most recently cellular Cu²⁺). How do these other mechanisms tie into those observed here regarding the lipid dependent electrostatic association of the CD with acidic membrane lipids? Increased ubiquitination of PD-L1 was observed for the PD-L1 3RE mutant, was this degradation SPOP dependent? Does it require Cu²⁺? Is the proposed lysosomal dependent mechanism via HIP1R at play at all here?

5) The authors' discuss palmitoylation of the CD. What is the contribution of palmitoylation to stability relative to that of the electrostatic association? Is electrostatic association with the membrane required for palmitoylation to occur? The 3RE mutant loses binding even with palmitoylation sequence present, suggesting electrostatics may contribute more. However a very strong charge clash was introduced by mutating all three residues to glutamic acid, which may prevent the tail from coming anywhere near the membrane. It would be interesting to look at a neutral CD mutant with the expectation that it would still lose electrostatic membrane association but possibly retain palmitoylation (ex. comparing to a C272S mutant).

6) The authors should perform a phylogenetic analysis (i.e., sequence analysis) to determine the conservation of the three arginine residues (or other charged groups) in the CD. The observation of conservation would support the wide spread utilization of the proposed mechanism.

7) The cytoplasmic tail also contains a number of acidic residues. It seems surprising that there was no evidence of salt bridge formation or stable secondary structure observed in the NMR analysis of the PD-L1 CD. Is there evidence of salt bridge formation in the absence of bicelles or if the bicelles are loaded with neutral lipids only? What happens to PD-L1 stability if the acidic residues are removed?

8) In figure 3C/D, only the CD domain is being expressed as a GFP fusion. Does any short basic peptide fused to GFP show membrane localization?

9) The authors justify the use of Metformin based on a few examples where this drug suppressed tumors at high doses and the fact that it contains a positively charged guanidinium moiety similar to that of arginine. In this study, millimolar concentrations of metformin were required to see a loss of PD-L1 membrane association, which seems non-specific and perhaps not the best strategy for targeting PD-L1 for degradation. Some direct comment on the doses used in the present studies and those used clinically would be useful. Are there better ways to prevent this association other than simply masking the charge? Does the effect observed with Metformin persist? Once PD-L1 is degraded how long does it take for PD-L1 levels to be restored?

Minor Points

It was difficult to follow in which experiments full-length PD-L1 was being expressed versus just the CD. This should be made more explicit as it is critical to how the data are interpreted.

The authors should be more explicit in their description of how the FRET assays were performed and analyzed, especially since the observed differences in photo bleaching are not always obvious from the confocal images.

The authors' refer to "we" in the introduction but it wasn't clear that their group published the referenced papers (19 and 20).

Reviewer #3 (Remarks to the Author):

In the present paper, Wen et al. have demonstrated that electrostatic interaction of the cytoplasmic domain of PD-L1 regulates the surface expression level of PD-L1 and suggest that metformin may exert its anti-tumorigenic effect by inhibiting the electrostatic interaction of PD-L1 with acidic phospholipids. The importance of electrostatic interaction of cytoplasmic unstructured chains and the plasma membrane was reported by the authors in 2008. The present paper is novel and interesting in that this hydrostatic interaction is associated with the ubiquitin-mediated degradation. The data are presented in an organized and clear way. However, the interaction was studied in RKO cells and the degradation was studied in HEK293 cells, raising a question whether these two observations could be associated each other. Moreover, there are some discrepancies between the presented images and quantified data. The comments are divided into major and minor; however, the minor questions may be more serious in that the reliability of each data is questioned in such seemingly trivial questions.

Major comments:

1. The localization of the PD-L1-CD3RE mutant is markedly different between HEK293 cells and RKO cells (Fig. 1D and 1E). Therefore, the proposal inferred from the observation in RKO cells could not be directly associated to HEK293 cells. The data presented in Fig. 4 should be repeated by using RKO cells.
2. The FRET measurement is based on the protocol reported previously by the authors (Ref. 24, 26). In these studies, the highest FRET values are approximately 50%, suggesting that this is the highest value obtainable by the distance effect. Further increase in FRET efficiency might be acquired by the increased R18 concentration on the membrane caused by PS. Probably, the most convincing data could be obtained by time lapse images as reported in ref. 26. We anticipate increase in FRET without changes of R18 fluorescence by the addition of PS.
3. Related to comment 2, the CD3 mutant with 3 amino-acid linker used in ref 26 should be tested for the effect of PS.
4. Fig. 5C: The effect of metformin should be shown by time lapse image. If the effect is mediated by the perturbation of electrostatic interaction, the effect should be faster than the effect that could be mediated by the inhibition of mitochondria.

Minor comments:

1. Fig. 1D, upper right panel, Fig. 3E, left panel, and Fig. 5C, right panel: In the mTFP image, there are some aggregates within the cytoplasm, which do not colocalize with R18. Nevertheless, the mTFP signal is increased after photo bleaching. Authors previously said "The reduction in donor fluorescence was only observed at the plasma membrane where TFP and R18 were colocalized, but not in intracellular compartments that only had a TFP signal" (ref. 26). This is not the case in the figures presented in this paper.

2. Fig. 3E, right panel: mTFP fluorescence of the 3RE mutant is apparently decreased after photobleaching. Similar decrease in mTFP fluorescence was observed in Fig. 5C. Why?

3. Fig. 4C: The WT PD-L1 was increased after 6 hours in all the three independent experiments. Therefore, this result could not be overlooked. If this is caused by decay of CHX, we will expect rapid recovery of the 3RE mutant also. But it does not happen. Why?

4. Statistical analyses are properly conducted. But, please make sure what independent experiment means. Moreover, N=3 samples should be shown with dots.

RESPONSES to REVIEWER COMMENTS

Reviewer #1 (Remarks to the Author):

This paper by Wen et al. describes an important scientific question on the stability of PD-L1, through the interaction of its cytoplasmic tail with the membrane. Though the results could be potentially interesting, the paper as such has some fundamental problems. The one at the top of the list is the structure.

The authors have solved the structure of the isolated cytoplasmic tail in POPG and report a structure with an RMSD of 0.5 Å. What is the meaning of this structure? Are the authors trying to say that this is the structure of the cytoplasmic tail in a biological context?

Answer: Thank you for this question. It has been reported that lipids in membranes are involved in signal transductions by interacting with receptor proteins in membranes. Although cytoplasmic tails of many receptor proteins behave as unstructured polypeptides in solution, some display certain structural features when bound to membranes, leading to identification of the membrane-interacting residues, and the role of the interactions in signal transductions (Xu et al., 2008; Yang et al., 2017).

The structure here is an experimentally derived representation of the tail in the context of a lipid bilayer. We believe this representation should reflect its form in the cell as it is consistent with mutagenesis and functional studies in cells.

The RMSD value reported only refers to the short stretch proximal to the TMD (260–275) which is well structured with rather stable interactions to lipid bilayers. The rest of the cytoplasmic domain is very dynamic and mobile. This situation is reminiscent of the CD3 ITAM (Xu et al., 2008). We have modified the related text in the revised manuscript (Page 5) to make this statement clear.

The structure looks very odd, it has a trapezium-shaped backbone at places. I am not sure if this will fit to any of Ramachandran parameters, just because we can obtain a tight structure to fit.

Answer: The stick diagram we used to display the structure in the original Figure 2 gave the wrong impression that the structure contains trapezium-shaped backbones. In the revised Figure 2, we used a ribbon diagram to show the structure. We also obtained the Ramachandran plot statistics of the structure. As shown in the revised Table 1, all the residues in the structure are located in either most favorable or additional allowed regions in the Ramachandran plot. Another

reason for the weird shape may be the angle of the view. For example, we don't observe the sharp angle at C272 after we rotated 180 degree along X axis (Fig. R1).

Figure R1 The NMR structure of CD₂₆₀₋₂₉₀ in lipid bilayers. (A) Ribbon diagram of a representative structure of CD₂₆₀₋₂₉₀ shallowly embedded in one leaflet of a lipid bilayer. The side chains of the three arginine residues and Cys272 for palmitoylation are shown. (B) The same representative structure of CD₂₆₀₋₂₉₀ viewed from the side (left) and rotated 180 degree along X axis (right).

Critical information as to how many long-range NOEs (i to i+4 and above) is missing.

Answer: We have listed the detailed NOE types including the long-range NOEs in Table 1.

I would request the authors to look at figure 2D and figure 2E, which are right below one another. Figure 2D shows a tight bundle of structures and figure 2E the hetero-NOE data shows that the section is for which they reported a tight bundle of structures (2D) is indeed highly dynamic. How do they reconcile these opposing results?

Answer: Although the overall structure of CD₂₆₀₋₂₉₀ in the presence of bicelles is disordered, the NMR spectral data obtained from the N-terminal 260-275 residues can be fit with a tight bundle of converged structural models. The conformational dynamics of this structural ensemble is most likely restricted by the bicelles, since the hetero-NOEs for the same N-terminal residues of CD₂₆₀₋₂₉₀ in the absence of bicelles are smaller and more dynamic. The comparison of hetero-NOEs for CD₂₆₀₋₂₉₀ in solution to those in the bicelles has been added to the revised Figure 2. The related text in page 5 of the revised manuscript was changed to: "The $\{^1\text{H}\} -^{15}\text{N}$ HetNOE values of the N-terminal 260-275 residues of CD₂₆₀₋₂₉₀ from the samples with bicelles are more positive and less dynamic than those from the samples without bicelles, indicating the conformation of the N-terminal half of CD₂₆₀₋₂₉₀ is restricted by the protein-membrane interaction (Fig. 2E)." Further, we note that the spread of NMR structures is determined by structural restraints and the temperature used in XPLOR calculation; it does not reflect protein dynamics in a quantitative manner.

The backbone structure in figure 2F right after R265 looks very unusual. What drives such an unusual structure? This square/trapezium-shaped backbone is seen again in the C-terminal. This is a serious problem that needs to be addressed.

Answer: As we mentioned above, the stick mode and the view angle made the figure look odd (Fig. R1). And we have changed Figure 2F into the cartoon mode.

I would suggest removing the structure and keeping the NMR data. The authors have to update their structure-calculation table with the number of long-range and short-range NOEs. They claim they have used dihedral restraints from TALOS. TALOS calculates dihedral restraints using chemical shift data that have been correlated to secondary structures and here there is no secondary structure. How can TALOS constraints be used?

Answer: We agree that it is generally difficult to represent the structure of a highly dynamic protein. However, we'd like to show a structural representation of the PD-L1-CD in the context of lipid bilayer to facilitate the overall presentation of our findings. We have added a sentence to caution the readers that the CD is highly dynamic and only structured in 260-275 regions (Page 5). Hence, the representation in Fig. 2 should be taken with caution.

NMR chemical shifts in proteins depend strongly on local structure. TALOS+ is an enhanced version of the earlier TALOS system, which adds a neural network classification scheme to improve the original TALOS database mining approach. The neural network analyzes the chemical shifts and sequence to estimate the likelihood of a given residue being in a sheet, helix, or loop conformation. Here, the backbone chemical shifts of PD-L1-CD were input into TALOS+ program, and the dihedral angle restraints derived from TALOS+ showed an extended conformation in PD-L1-CD, no sheet or helix secondary structure formed, which is also supported by our NOE pattern.

It would have been prudent to have solved the structure with the transmembrane helix, which will position the cytoplasmic tail in the membrane. Without that this structure holds no meaning as presented.

Answer: It's true that solving a structure with both the cytoplasmic tail and the transmembrane helix would provide a whole picture for the membrane association of PD-L1. However, in our study the cytoplasmic tail alone associates with a membrane displaying an interesting structural transition from the tail in solution, with an important implication to PD-L1 stability and function in cancer cells. Previous structural studies of CD3 and CD28 cytoplasmic tails without transmembrane helix also established their roles in signal transduction across cellular membranes (Xu et al., 2008; Yang et al., 2017). We will try to solve the CD structure with the transmembrane helix, but it would belong to a future project.

Other Concerns.

a) Fig1. The authors should provide a plot of chemical shift perturbations as well as peak intensities plotted against the protein sequence. If the claim that the cytoplasmic tail is getting embedded in the membrane (DMPG) with a well-defined structure is correct, then they should observe a remarkable reduction in intensities.

Answer: Good suggestion! We have made the plot of peak intensities against the protein sequence in Figure S2, which shows a remarkable reduction in intensities of N-terminal residues consistent with the chemical shift perturbations.

b) The spectrum of the cytoplasmic tail in solution seems to have remarkable dispersion compared to the DMPG spectrum, especially look at the peak at ^1H -7.7ppm and ^{15}N -121 ppm. It is missing from the DMPG spectrum. What is the explanation for this? Why does it reappear in the POPG spectrum?

Answer: The spectra overlapped very well at the peak ^1H -7.7ppm and ^{15}N -121 ppm in Figure 1B, so this peak of the DMPG spectrum was covered by the peak from the in solution spectrum, but it is always there as we can see it from other figures, such as Figure 5. We have separated the spectra in Figure R2 to show the covered peaks.

Figure R2 Superimposed ^1H - ^{15}N TROSY-HSQC spectra (A) of $\text{CD}_{260-290}$ in solution (blue), DMPG/DHPC bicelles (red) and in DMPC/DHPC bicelles (green) and their separated spectra (B-D).

c) One problem with the experiment is the DMPG/DHPC mix used for bicelles. All the acidic lipids will cluster on the low curvature sides of the bicelles likely providing an effective 100% acidic surface, far from plasma membrane acidity. It would be better to do the experiment with 25%DMPG-75%DMPC/DHPC bicelles. Or even better, a POPG/POPC mix. This is absolutely necessary to validate that the structure determination is close to that inside cells and not an artifact of the artificially high acidic lipid content.

Answer: This is a great question! We have compared the spectrum in DMPG/DHPC bicelles with

the spectrum in POPG/DHPC bicelles, they overlapped very well (Figure S2C). We have collected the spectra at different ratio of DMPG/DMPC (Figure S2A), and the chemical shifts of the resonances moved continuously from the in-solution state to the membrane-bound state, indicating an equilibrium exists between the non-membrane-bound state and membrane-bound state. We agree that it would be better to do the experiment in a more “cell relevant” model membrane, 25%DMPG-75%DMPC/DHPC bicelles. However, to simplify the experimental setup and data analysis, we chose 100% DMPG/DHPC to obtain the maximum membrane-bound form of PD-L1-CD to show how the cytoplasmic tail interacting with the membrane.

d) Another potential problem with that experiment is the extremely low ionic strength of the buffer (25 mM MES, no salt) which is also far from cell conditions. A control experiment with 100-150 mM NaCl should be performed to see if the electrostatic interaction still takes place.

Answer: We have done the control experiment with 100 mM NaCl (Fig. R3) and the spectrum is very similar to that with no salt.

Figure R3 Superimposed ^1H - ^{15}N TROSY-HSQC spectra of CD₂₆₀₋₂₉₀ in DMPG/DHPC bicelles in the absence of NaCl (red) and in the presence of 100 mM NaCl (blue).

e) Was there a reason to change from DMPG to POPG for structure determination?

Answer: We have added the comparison of the spectra in DMPG/DHPC bicelles and in POPG/DHPC bicelles (Figure S2C), they are almost the same. We used the POPG bicelles to better mimic cell membranes and to detect the membrane insertion of CD₂₆₀₋₂₉₀. We used deuterated DMPG for the experiments requiring deuterated lipids, since deuterated POPG is more expensive.

f) It was not very clear to me what secondary structure may or may not form upon lipid binding. A lot of similar membrane interacting protein form imperfect alpha-helices but it is not the case here? The authors could compare with the structure of alpha-synuclein bound to lipids (Ulmer et al 2005 JBC 280).

Answer: It was reported that membrane interacting proteins can form imperfect alpha-helices (Xu et al., 2008), as the reviewer mentioned, alpha-synuclein is predominantly a random coil in

aqueous solution but adopts helical secondary structure upon association with negatively charged detergents (Ulmer et al., 2005). But in our case, no obvious secondary structure form upon binding to lipid bicelles, which is similar to the results observed in CD28 cytoplasmic tail (Yang et al., 2017). We have added this comparison in the discussion (Page 8).

g) The authors should discuss the effect of calcium ions if anything has been reported. Influx of calcium is likely to disrupt membrane interactions by shielding acidic lipids as well (Zhang et al 2014 BBA 1838).

Answer: In T cells, the calcium influx does efficiently regulate the interaction of the CD3 ϵ and CD3 ζ tails with the membrane. The calcium influx was not reported in cancer cells, however, a high dietary calcium intake was studied in some cancers, for example, the risk of colorectal cancer and rectal cancer has been found to be reduced by a high calcium intake (Han et al., 2015; Wu et al., 2002), whereas the risk of prostate cancer increased (Wilson et al., 2015). Therefore, the calcium effects on PD-L1 is unclear or we guess calcium disruption on PD-L1-membrane interactions is dependent on cancer types. We have added the discussion about the effect of calcium ions in the discussion part (Page 9).

The literature we found about Zhang et al 2014 BBA 1838 (Zhang et al., 2014) reported that a cyclic anionic lipopeptide, daptomycin, underwent Ca²⁺-dependent structural transitions and the presence of negatively charged lipids allowed daptomycin to insert into and perturb bilayer membranes with acidic character, instead of positive charged residues in the peptide interacting with the anionic lipids and calcium disrupting the electrostatic interactions. We prefer not to add this into the discussion. If this is not the same reference as the reviewer mentioned, we would like to know the details about the correct reference and add into our discussion.

h) The authors state "A stock solution of 2 M metformin was added to the NMR sample at the molar ratios of [metformin]:[DMPG]= 0, 0.05, 0.1, 0.2, 0.4, 0.6, 1.0, 2.0. " Is this not a very high metformin concentration? Is this even biologically relevant? If we want to connect to the biology that they are implying in the discussion section, we should limit our discussions of results to metformin concentrations of ~ 40uM.

Answer: To avoid the NMR sample volume increase upon the addition of metformin, we used a high-concentration of metformin stock solution, which gave the final metformin concentrations 1 mM to 40 mM. Considering the existence of free bicelles (mM range) and each bicelle can hold more than one metformin molecules, the efficient concentration on protein in bicelles was much lower than the added amount. The high metformin concentration used in our NMR titrations are within the biologically relevant range, as many previous studies showed that only millimolar concentrations of metformin led to a significant reduction of the cellular abundance of PD-L1 (Cha et al., 2018; Elgendy et al., 2019; Xue et al., 2019). And a number of clinical studies also used metformin in high doses of 1,500–2,250 mg per day to reduce the risk of cancer (Lee et al., 2019; Wang et al., 2019). Therefore high metformin concentrations are needed to show the inhibition effects *in vitro* and *in vivo*.

NMR experiments with arginine will be a welcome addition to the manuscript.

Answer: NMR experiments with arginine have been done and showed similar chemical shift changes as metformin (Fig. R4). However, our studies focus on the druggable compounds, so we prefer not to add the arginine data to the manuscript. If the reviewer insists, we would be happy to do that.

Figure R4 The effects of arginine on CD₂₆₀₋₂₉₀ in DMPG bicelles. (A) Superimposed ^1H - ^{15}N TROSY-HSQC spectra of CD₂₆₀₋₂₉₀ with DMPG bicelles (red), and different concentrations of arginine were titrated into the CD₂₆₀₋₂₉₀-DMPG bicelle at a molar ratio of arginine:[DMPG] from 0.05 to 2. (B) Chemical shift perturbations were quantified and plotted on a molar ratio of arginine:[DMPG]=2.0.

i) The authors need to show a heteroNOE experiment in the non-bicelle (solution) state to show the difference due to membrane association.

Answer: We have done the hetero-NOE experiment in the non-bicelle-bound (solution) state and shown the results in the revised Figure 2E, which show that the membrane association significantly changes the dynamics of the N-terminal region of PD-L1-CD.

j) In the introduction the authors write “Besides the multiple regulatory roles of the cytoplasmic domain, several cancer derived mutations were found within this domain, including R260C, R262K, D276Y, and T290M, alluding its critical role in cancer cell survival” In the light of their studies what is effect of these mutations on membrane association and the stability of the PD-L1.

Answer: We thank this reviewer for this inspiration! We have added this into our discussion (Page 10) with the following sentences: “We note that some cancer derived mutations found in this cytoplasmic region might increase PD-L1 stability through enhancing the membrane association

to escape the immunosurveillance. For instance, D276Y and T290M can increase the residue hydrophobicity with the replacements and stabilize PD-L1 with stronger hydrophobic membrane interactions. R260C might provide a new position for the palmitoylation and therefore improve the stability. It's not clear whether R262K enhances the membrane interactions of PD-L1, since R has a guanidino group but a shorter acyl chain. Further studies of these mutations in biochemical and cell biological systems are required to confirm or refute the expectations."

Minor points.

1) Please consider working on the title. PD-L1 is repeated in a span of two words.

Answer: We have changed the title to "PD-L1 degradation is regulated by electrostatic membrane association of its cytoplasmic domain".

2) Equation 1 needs a definition of all the terms. What is tau? For those of us who are not familiar with this way of representing the data, the authors have to explain what PREamp means, especially in this case the higher number means PRE effect.

Answer: We have added the definition in Equation 1, in which τ is the decay constant. The explanation about PREamp has been added in the methods "Lipophilic PRE and solvent PRE analysis of membrane-bound PD-L1-CD" (Page 13). For more details, please see the reference "Optimal Bicelle Size q for Solution NMR Studies of the Protein Transmembrane Partition", Alessandro Piai, Qingshan Fu, Jyoti Dev, James J Chou, *Chemistry*, 23(6):1361-1367. doi: 10.1002/chem.201604206.

3) Figure 5, the CSP has to be plotted as a graph for each residue so that we can clearly see what is happening.

Answer: As suggested, we have made a CSP plotted graph to Figure 5.

4) What is the saturation time in the heteroNOE experiment?

Answer: We have added more details about heteroNOE experiment into the paper's methods (Page 12).

5) The authors state "Protein intramolecular distance restraints for membrane bound PD L1 CD structure calculations were derived from cross peaks in a simultaneous ^{15}N and ^{13}C NOESY HSQC (τ NOE= 120)". What is a simultaneous NOESY? Are they referring to time-shared NOESY? Is the reference (29) the correct one here?

Answer: It is also a time-shared NOESY. Please refer to Pascal S.M., Muhandiram D.R., Yamazaki T., Forman-Kay J.D., Kay L.E. 1994. Simultaneous acquisition of ^{15}N -edited and ^{13}C -edited NOE spectra of proteins dissolved in H_2O . *J. Magn Reson B* 103:197-201. We have changed to the correct reference.

6) The description of the NMR method is very minimal and should be expanded to contain details of the 3D experiments, how many points were collected, what mixing times were used for the TOCSY experiment, how was the data processed etc.

Answer: We have added supplementary table S1 to show the detailed description of the NMR experiments.

7) They used a deuterated sample for backbone experiment, while a protonated sample for the side-chain experiment, which is more demanding in terms of relaxation. Considering the protein is very flexible, was the deuteration actually needed in the first place.

Answer: We agree that the deuteration is not necessary, but we worked on the bicelle system with a large q value to resemble the native membranes, so we used the deuterated sample to get better signals.

Reviewer #2 (Remarks to the Author):

Electrostatic membrane association of the cytoplasmic domain of PD-L1 regulates PD-L1 degradation.

Maorong Wen et al.

Reviewer Summary

The authors employ a combination of NMR techniques, in vitro assays and FRET confocal imaging to investigate the regulation of PD-L1 stability via interaction of the PD-L1 cytoplasmic tail domain (CD) with plasma membrane lipids. Their structural analysis of the PD-L1 cytoplasmic tail (residues 260 - 290) bound to lipid bicelles identified three specific arginine residues (R260, R262 and R265) that associate directly with acidic lipids in the membrane. Using CD mutants of PD-L1 they demonstrate that loss of membrane association results in less stable PD-L1 that has higher levels of ubiquitination and is consequently degraded faster inside cells. Additionally, they show that the FDA-approved drug Metformin, which contains a positively charged guanidinium group, can block membrane association of WT PD-L1-CD and subsequently decrease PD-L1 protein levels (total and cell-surface expressed). Together their data suggest a novel mechanism by which membrane association of the cytoplasmic domain of PD-L1 regulates PD-L1 turnover, thereby controlling cell-surface levels of PD-L1. Overall the experiments are well designed, well presented and the results are internally consistent with each other. However, the manuscript would benefit from additional consideration of how electrostatic association between the CD and membrane lipids relates to previously identified mechanisms that regulate PD-L1 turnover (SPOP dependent proteasome degradation, Cu^{2+} influx etc.) and also what role (if any) PD-1 or B7-1 engagement may play in this process. Detailed comments follow below:

Reviewer Comments and Suggestions

1) This manuscript does not consider what role receptor engagement might play in regulating the cytoplasmic domain even though PD-1 engagement drives the immunosuppressive capacity of tumor cells. Residue R260 in PD-L1 is located right at the C-terminal end of the transmembrane helix; in fact prediction algorithms place this residue within the transmembrane domain (239 – 261). This suggests the possibility that PD-1 engagement might trigger repositioning of the cytoplasmic tail domain via a mechanically transduced conformational change translated through the transmembrane domain. The authors could directly address potential PD-1 dependent effects using their in vitro FRET and HEK cell expression experiments, both of which utilized expression of full-length PD-L1 constructs.

Answer: Good point! We have performed the FRET experiment and found that FRET efficiency and the protein levels of PD-L1 increased after PD-1 bound to PD-L1, indicating that PD-1 engagement enhances PD-L1-CD association with the membrane and PD-L1 stability (Fig. R5). We agree that this repositioning may be introduced via the mechanically conformational change through the transmembrane (TM) domain. However, the detailed mechanisms of the signal transductions facilitated by TM are incompletely understood. The study of structure and signal transduction of transmembrane domain of PD-L1 is currently under way. Our primary aim of our manuscript is to identify the interaction of PD-L1-CD with membranes and the consequence on PD-L1 level in cell membranes, we would prefer to address this question in our next paper.

Figure R5 PD-1 enhances PD-L1-CD bound to the membrane. (A) Schematic model of PD-1 bound to PD-L1 enhancing the membrane association of PD-L1-CD. (B) Western bolt analysis of PD-L1 levels in RKO cells after treated by PD-1 for different times. (C) Dequenching FRET assay showed the enhanced effect of PD-1 on PD-L1-CD-membrane interaction. (D) The statistics analysis of the dequenching FRET efficiencies from C. Each black symbol represents one individual cell; the bars represent the mean \pm S.E.M.; n = 25. Scale bar, 5 μ m.

2) PD-L1 also interacts with CD80 in cis and formation of the cis complex has been shown to inhibit PD-1 activity in vivo. Although CD80 is most commonly down regulated in solid tumors, both on tumor cells and on tumor associated macrophages, it is possible that in the context of normal APC cells, B7-1 binding in cis affects overall PD-L1 stability. Both CD80 and CD86 also have short cytoplasmic domains that are highly basic (pI of 11.47 and 9.72 respectively). Do the authors suggest that this is a general mechanism for all of the B7 family ligands? Does the overall charge of the cytoplasmic domain correlate with stability?

Answer: Thank you for this question! We agree it could be a general mechanism for the B7 family ligands considering their highly basic cytoplasmic domains. Previous studies on T cell receptors reported that a basic residue rich stretch (BRS) in CD3 ϵ cytoplasmic tail binds to the plasma membrane and its dissociation from the membrane is required for its phosphorylation (Xu et al., 2008). Later studies further showed that BRS motifs within the cytoplasmic tails of CD28 (Yang et

al., 2017), CD3 ζ (DeFord-Watts et al., 2011) and TCR ζ (Zhang et al., 2011) mediate association with the plasma membrane and thus regulate the downstream activity. Moreover, the roles of C-terminal membrane-proximal basic residues were found to be required for activation, deactivation, degradation, and/or endocytosis of other plasma membrane proteins including receptors (EGFR, GPCRs), channels and kinases (Li et al., 2014; Okamoto et al., 2013; Tetsuka et al., 2004). Although the juxtamembrane polybasic regions conserved in other B7 family ligands have not been studied yet, that the interactions between the polybasic residues and acidic lipids playing an important role in functional regulations may be a general mechanism. We have added this into our discussion (Page 10).

The overall charge of a cytoplasmic domain is expected to correlate with its stability, since the more basic the cytoplasmic domain is, the more tightly it binds to the membrane. However, our NMR experiments and mutagenesis data of 2KE (K270E/K271E) showed that K270 and K271 in the cytoplasmic domain are not essential for membrane interaction (Fig. S3C-D), so the position of the polybasic residues in the cytoplasmic domain also matters. Besides the charge and position, we believe there are some other factors involved in the protein stability, for example, hydrophobic interactions and protein palmitoylation.

3) The cytoplasmic domain of PD-L2 is even more basic than that of PD-L1 (pI of 10.30 compared to 9.52 for PD-L1). The authors' hypothesis would suggest that PD-L2 should be more membrane associated and therefore more stable than PD-L1. These features might provide another mechanism for driving specific PD-L1 and PD-L2 functionality.

Answer: Great idea! It will be interesting to see the difference between PD-L1 and PD-L2. And we would like to perform the study of PD-L2 cytoplasmic domain to identify the specific mechanism for PD-L2 in the near future.

4) There are a number of other mechanisms that have been shown to play a role in regulating PD-L1 stability and turnover (i.e. SPOP binding, CSN5 binding, HIP1R binding, most recently cellular Cu²⁺). How do these other mechanisms tie into those observed here regarding the lipid dependent electrostatic association of the CD with acidic membrane lipids? Increased ubiquitination of PD-L1 was observed for the PD-L1 3RE mutant, was this degradation SPOP dependent? Does it require Cu²⁺? Is the proposed lysosomal dependent mechanism via HIP1R at play at all here?

Answer: This is a great question! We think the lipid dependent electrostatic association are related to some of the downstream regulations on PD-L1 stability, though further investigations are needed to figure out which one or more of the pathways are involved. Previous studies reported that SPOP and HIP1R bind to the cytoplasmic tail of PD-L1, so we believe that the release of PD-L1-CD from the membrane facilitate the binding to SPOP and HIP1R and therefore enhance the degradation. In addition, the membrane association of PD-L1-CD disrupts palmitoylation of the CD. The consequences for the SPOP, HIP1R binding and palmitoylation disruption are the same, i.e., decreasing PD-L1 stability and reducing protein levels. However, whether the degradation of PD-L1^{3RE} mutant is SPOP dependent has not been studied. And, only

further experiments can exactly pin down the involvement of membrane association in different regulation pathways. We have added these comments to the revised Discussion (Page 9).

CSN5 deubiquitinates PD-L1 for protein stabilization (Lim et al., 2016). We are not sure whether the ubiquitinated PD-L1 maintains the membrane association or not. We guess that the membrane interaction of PD-L1-CD is somehow affected by the ubiquitination, and CSN5 binding to PD-L1-CD is the competitive outcome of membrane interaction and ubiquitination.

It was reported that cellular Cu^{2+} modulates oxidative phosphorylation in tumors and activates NF κ B, JAK/STAT and PI3K/Akt/mTOR/S6K signaling, thus triggering PD-L1 upregulation (Voli et al., 2020). We have titrated Cu^{2+} to PD-L1-CD (Fig. R6), interestingly, we found Cu^{2+} caused peak intensities of the N-terminal residues bound to the membrane decreasing significantly at a low concentration (0.5 mM), suggesting these residues may aggregate or embed into the membrane deeper. In contrast, other cations including Ca^{2+} , Mg^{2+} , Zn^{2+} didn't decrease the peak intensities even at a much higher concentration (5 mM), but caused chemical shift perturbations at different extents (Fig. R6). So Cu^{2+} behaved differently from other +2 cations, we guess that its effects on PD-L1 are not purely based on its charges, the detailed mechanism of Cu^{2+} on PD-L1 needs further investigations.

Figure R6 Superimposed ^1H - ^{15}N TROSY-HSQC spectra of CD₂₆₀₋₂₉₀ in DMPG/DHPC bicelle (red), and in the presence of 5 mM Ca^{2+} (black) (A), 5 mM Mg^{2+} (green) (B), 5 mM Zn^{2+} (dark blue) (C) or 0.5 mM Cu^{2+} (light blue).

5) The authors discuss palmitoylation of the CD. What is the contribution of palmitoylation to stability relative to that of the electrostatic association? Is electrostatic association with the membrane required for palmitoylation to occur? The 3RE mutant loses binding even with

palmitoylation sequence present, suggesting electrostatics may contribute more. However a very strong charge clash was introduced by mutating all three residues to glutamic acid, which may prevent the tail from coming anywhere near the membrane. It would be interesting to look at a neutral CD mutant with the expectation that it would still lose electrostatic membrane association but possibly retain palmitoylation (ex. comparing to a C272S mutant).

Answer: PD-L1 is palmitoylated at C272 by DHHC3 acetyltransferase (Yao et al., 2019), a homologue to DHHC20 and DHHC15. As indicated from the DHHC20 and DHHC15 crystal structures (Rana et al., 2018), the substrate palmitoyl-CoA locates in the DHHC cavity within the membrane (Fig. R7A). Our NMR results showed that the polybasic residues enhance the membrane association to embed C272 into the membrane, so that the palmitoylation site is close to the substrate and thus facilitates PD-L1 palmitoylation. We compared the membrane interaction of cytoplasmic tail of PD-L1 for C272S, 3RA (neutral CD mutant) and 3RE mutants. The results (Fig. 3E and Fig. R7B) show that both 3RA and 3RE mutants can significantly decrease the cytoplasmic tail binding to the membrane, C272S mutant also decreased the membrane binding but in a smaller extent. The results suggested that the polybasic residues in the juxtamembrane domain are very important to the membrane interaction.

Figure R7 (A) Molecular surface rendition of the DHHC enzyme with the substrate acyl chain shown in yellow and red spheres. The putative direction of substrate approach is shown with a cyan arrow. (B) De-quenching FRET measurements showed the membrane interactions of PD-L1-CD for wildtype and mutants. In the bottom graph, each black symbol represents one individual cell; the bars represent the mean \pm s.e.m; $n = 25, 25, 12$ and 22 for PD-L1 wildtype, PD-L1^{C272S}, PD-L1^{3RA} (R260A/R262A/R265A) or PD-L1^{3RE} (R260E/R262E/R265E), respectively. Scale bar, $5 \mu\text{m}$.

6) The authors should perform a phylogenetic analysis (i.e., sequence analysis) to determine the conservation of the three arginine residues (or other charged groups) in the CD. The observation of conservation would support the wide spread utilization of the proposed mechanism.

Answer: We have added a phylogenetic analysis as Figure S3B. The poly-basic residues in the

juxtamembrane is conserved (Fig. S3A).

7) The cytoplasmic tail also contains a number of acidic residues. It seems surprising that there was no evidence of salt bridge formation or stable secondary structure observed in the NMR analysis of the PD-L1 CD. Is there evidence of salt bridge formation in the absence of bicelles or if the bicelles are loaded with neutral lipids only? What happens to PD-L1 stability if the acidic residues are removed?

Answer: In our NMR experiments, we found only the N-terminal region of cytoplasmic tail of PD-L1 can interact with the membrane, the C-terminal region is very dynamic and mobile on the surface of the membrane and mostly not associated with the membrane. The NMR results showed PD-L1-CD in the absence of bicelles or in the neutral lipids have very similar spectra (Fig. 1B) with unstructured and dynamic structures. No evidences showed the salt bridges formation in the absence of bicelles (No NOEs observed between these residues). Though these acidic residues didn't interact with the polybasic residues in the N-terminus, it was reported that they are essential to recognize SPOP binding (Zhang et al., 2018), which in turn affects PD-L1 ubiquitination and stability.

8) In figure 3C/D, only the CD domain is being expressed as a GFP fusion. Does any short basic peptide fused to GFP show membrane localization?

Answer: Many peptides can change the GFP location to cell membranes. For example, a group of short, highly basic peptides can transfer the GFP protein into cells (Kadkhodayan et al., 2016); a membrane GFP line generated using amino-terminal myristoylation/palmitoylation sequence to GFP is a powerful tool for tracking cell movements, cell boundaries, cell lineage and cell shape changes *in vivo* and has provided insights on primitive steak formation (Rozbicki et al., 2015) and early embryonic development (Ferro et al., 2020).

9) The authors justify the use of Metformin based on a few examples where this drug suppressed tumors at high doses and the fact that it contains a positively charged guanidinium moiety similar to that of arginine. In this study, millimolar concentrations of metformin were required to see a loss of PD-L1 membrane association, which seems non-specific and perhaps not the best strategy for targeting PD-L1 for degradation. Some direct comment on the dose used in the present studies and those used clinically would be useful. Are there better ways to prevent this association other than simply masking the charge? Does the effect observed with Metformin persist? Once PD-L1 is degraded how long does it take for PD-L1 levels to be restored?

Answer: Good point! We have added the comments about the high doses used in other cell studies and clinical studies in the discussion part (Page 10). We agree that the disruption by metformin in our study is mainly due to a nonspecific effect and more specific molecules targeting on PD-L1-CD-membrane interactions should be designed, which requires further drug development.

The effect caused by metformin can last at least 72 hours (Fig. R8A) in the presence of metformin

in cell cultures. After we removed metformin in the medium, PD-L1 levels recovered in 48 hours and increased significantly in 72 hours (Fig. R8B).

Figure R8 The persistence of metformin effects on PD-L1. RKO cells treated with 5 mM metformin for 12 hours (C as control), continued culturing the cells in the medium with (A) or without (B) metformin for extra 3 days. Western blot analysis for the PD-L1 level in the cells.

Minor Points

It was difficult to follow in which experiments full-length PD-L1 was being expressed versus just the CD. This should be made more explicit as it is critical to how the data are interpreted.

Answer: We apologize for the confusion. We have kept PD-L1-CD for “the cytoplasmic domain of PD-L1” when we expressed full-length PD-L1 to observe its behavior on the cytoplasmic domain, and used CD₂₆₀₋₂₉₀ specifically referring to the human short peptide (only containing residues 260-290) used in all NMR experiments and confocal microscopy experiments in Figure 3.

The authors should be more explicit in their description of how the FRET assays were performed and analyzed, especially since the observed differences in photo bleaching are not always obvious from the confocal images.

Answer: We have added more details about the FRET assays in Page 14-15.

The authors’ refer to “we” in the introduction but it wasn’t clear that their group published the referenced papers (19 and 20).

Answer: We have removed “we noticed that”

Reviewer #3 (Remarks to the Author):

In the present paper, Wen et al. have demonstrated that electrostatic interaction of the cytoplasmic domain of PD-L1 regulates the surface expression level of PD-L1 and suggest that metformin may exert its anti-tumorigenic effect by inhibiting the electrostatic interaction of PD-L1 with acidic phospholipids. The importance of electrostatic interaction of cytoplasmic unstructured chains and the plasma membrane was reported by the authors in 2008. The present paper is novel and interesting in that this hydrostatic interaction is associated with the ubiquitin-mediated degradation. The data are presented in an organized and clear way. However, the interaction was studied in RKO cells and the degradation was studied in HEK293 cells, raising a question whether these two observations could be associated each other. Moreover, there are some discrepancies between the presented images and quantified data. The comments are divided into major and minor; however, the minor questions may be more serious in that the reliability of each data is questioned in such seemingly trivial questions.

Major comments:

1. The localization of the PD-L1-CD3RE mutant is markedly different between HEK293 cells and RKO cells (Fig. 1D and 1E). Therefore, the proposal inferred from the observation in RKO cells could not be directly associated to HEK293 cells. The data presented in Fig. 4 should be repeated by using RKO cells.

Answer: We think the significant differences for the localization of PD-L1-CD^{3RE} mutant in HEK293 cells and RKO cells mentioned by the reviewer are actually referred to Fig. 3D (HEK293) and Fig. 3E (RKO), which are caused by the different constructs used in the experiments. In Figure 3D we used a construct only containing the cytoplasmic domain part (CD₂₆₀₋₂₉₀, residues 260-290) attached to GFP, when the three arginines were mutated to glutamic acids, the peptide CD₂₆₀₋₂₉₀^{3RE} can't transfer GFP to the cell membrane, so plenty of the GFP proteins expressed in the cytosol. While in Figure 3E, we used the full length PD-L1 which has a transmembrane domain, so even with the mutations, the mutant proteins mainly expressed in the membrane. We have changed the name for the construct to avoid the confusion.

As suggested, we have repeated the CHX-chase assay and ubiquitination detection on PD-L1^{WT} and PD-L1^{3RE} in RKO cells, showing similar results as HEK293 cells. The new data have been added to Figure S6.

2. The FRET measurement is based on the protocol reported previously by the authors (Ref. 24, 26). In these studies, the highest FRET values are approximately 50%, suggesting that this is the highest value obtainable by the distance effect. Further increase in FRET efficiency might be acquired by the increased R18 concentration on the membrane caused by PS. Probably, the most convincing data could be obtained by time lapse images as reported in ref. 26. We anticipate increase in FRET without changes of R18 fluorescence by the addition of PS.

Answer: Thank you for this question! FRET values are dependent on the distance effect, the farther in distance, the less efficient for FRET. For example, when using 25 and 50 residue linkers, the FRET efficiencies exhibited lower values than 3 residue linker that correlated with the distance of the TFP domain from the plasma membrane. In Ref 26, the CD3 ϵ cytoplasmic domain is 57 residues in length with a distance of 200 Å from the membrane in a fully extended conformation its C terminus. In Ref 24, the CD28 has a 41-residue cytoplasmic domain, both of them are longer than PD-L1-CD (31 residues). On the other hand, FRET values are also dependent on how many proteins are attached to the membrane, more proteins attached to the membrane, higher transfer rates for the FRET. So the highest FRET values can be larger than 50%, in our case, it reaches 75%.

We have performed the time lapse FRET experiments with the addition of PS, showing increased FRET efficiencies. The results were added in Figure S3. We have also recorded the fluorescence of R18 in native RKO cells at different time points in the presence of PS, showing slight intensity decreases over time due to the dye photobleaching (Fig. R9), indicating no increases of R18 concentration on the membrane caused by PS. Therefore, the increased FRET efficiencies are not caused by the R18 concentration changes.

Figure R9 Time lapse imaging of R18 fluorescence in the RKO cells. The first fluorescence image was recorded for R18 pre-stained RKO cells, then the cells were treated with 5 μ M PS (related to Figure 1D) and the fluorescence images were recorded at 0.5, 1 and 2 hours. Scale bar, 5 μ m.

3. Related to comment 2, the CD3 mutant with 3 amino-acid linker used in ref 26 should be tested for the effect of PS.

Answer: This is a great suggestion! We have performed the FRET measurements of CD₂₆₀₋₂₉₀^{3RE} with 3 amino-acid linker in the presence of PS, showing that PS has little effect on the mutant. The results have been added in Figure 3E.

4. Fig. 5C: The effect of metformin should be shown by time lapse image. If the effect is mediated by the perturbation of electrostatic interaction, the effect should be faster than the effect that could be mediated by the inhibition of mitochondria.

Answer: Until now, the effect of metformin on mitochondria is still unknown. Paradoxical effects of metformin on mitochondrial respiratory chain activity have been published in the literature (Docrat et al., 2020; Fontaine, 2018; Vial et al., 2019; Wang et al., 2019). It is not clear how fast metformin inhibits the mitochondria. Usually, cells were treated with metformin 24 hours then observe the effects on mitochondrial metabolism (Wang et al., 2019). Here, we have recorded the time lapse fluorescence images of RKO cells expressed PD-L1-mTFP in DMSO or metformin

treated (Fig. R10A). The fluorescence intensities decreased in a short time (0.5 h). Measurement of FRET efficiency treated with metformin for different times (Fig. R10B) showed that after 1 h, FRET efficiencies reduced significantly for the cells.

Figure R10 FRET approach to examine metformin effects on PD-L1. (A) Time lapse fluorescence images of RKO cells expressed PD-L1-mTFP in DMSO or treated by 5 mM metformin. (B) FRET efficiencies were measured and plotted. RKO cells expressing PD-L1-mTFP were treated with 5 mM metformin for 0.5, 1, 2, and 4 h. The whisker-dot plot shows the statistical results of FRET efficiency, in which each dot represents the FRET efficiency from one individual cell; each whisker represents the mean \pm SD from $n = 19, 15, 17, 18$ and 34 from the 0, 0.5, 1, 2, and 4 h after metformin treated respectively. Scale bar, $5 \mu\text{m}$.

Minor comments:

1. Fig. 1D, upper right panel, Fig. 3E, left panel, and Fig. 5C, right panel: In the mTFP image, there are some aggregates within the cytoplasm, which do not colocalize with R18. Nevertheless, the mTFP signal is increased after photo bleaching. Authors previously said “The reduction in donor fluorescence was only observed at the plasma membrane where TFP and R18 were colocalized, but not in intracellular compartments that only had a TFP signal” (ref. 26). This is not the case in the figures presented in this paper.

Answer: Good points! The camera focus sometimes may drift a little bit during the image taken. For example, in the previous reference, the intracellular compartments sometimes also show higher intensities (Fig. R11A, top) sometimes show lower intensities (Fig. R11A, bottom). In our FRET images some cells kept the same intensities for the intracellular parts and showed intensity increases on the membrane (Fig. R11B).

We have adjusted the brightness and contrast of the images in the Figure 1D, 3E and 5D with the same setting in BP and AP (brightness+50, contrast+100) to give the same brightness for the intracellular intensities. One example was shown in Figure R11C, in which fluorescence intensities significantly increased at the plasma membrane after photobleaching with adjusted brightness and contrast.

Figure R11 Comparison of the fluorescence for intracellular compartments and plasma membranes. (A) FRET efficiencies of CD3ε measured in the reference (Xu et al., Cell 2008). (B) FRET efficiencies determined for RKO cells expressing PD-L1^{WT}-mTFP. The same experiment dataset as PD-L1^{WT}-mTFP in Figure 3E. (C) The adjusted images of PD-L1^{WT}-mTFP in Figure 3E. The membrane parts were zoomed in large (1) and the intracellular parts were zoomed in large (2). Scale bar, 10 μm.

2. Fig. 3E, right panel: mTFP fluorescence of the 3RE mutant is apparently decreased after photobleaching. Similar decrease in mTFP fluorescence was observed in Fig. 5C. Why?

Answer: We agree that some regions in the image (Fig. 5D) showed decreased mTFP fluorescence intensity (1 in Fig. R12A), but some regions increased the fluorescence (2 in Fig. R12A). Our

conclusion was drawn based on the statistics analysis of the total mTFP fluorescence, which showed slightly increased values after photobleaching (Fig. R12B) for the 3RE mutant (Fig. 3E) and metformin treated cells (Fig. 5D). The 3RE mutant disrupts the membrane association, therefore the mTFP fluorescence is not enhanced as much as the wild type.

Figure R12 mTFP and R18 fluorescence intensity before photobleaching (BP) and after photobleaching (AP). (A) Zoomed regions of mTFP fluorescence BP and AP, showing region 1 decreased while region 2 increased the intensities in AP. Scale bar, 10 μ m. (B) The total mTFP fluorescence and R18 fluorescence intensities were recorded separately three times before photobleaching (BP) and after photobleaching (AP) of PD-L1^{3RE} in Figure 3E and metformin treated PD-L1^{WT} in Figure 5D. mTFP fluorescence intensity is shown in cyan, R18 fluorescence intensity is shown in red.

3. Fig. 4C: The WT PD-L1 was increased after 6 hours in all the three independent experiments. Therefore, this result could not be overlooked. If this is caused by decay of CHX, we will expect rapid recovery of the 3RE mutant also. But it does not happen. Why?

Answer: We have checked the western blot results of PD-L1^{WT} and PD-L1^{3RE} from the three independent experiments (Fig. R13). It is true that the expression of PD-L1^{WT} recovered after 6 hours while PD-L1^{3RE} didn't show the same trend. However, from the expression decay curves, we could see the protein levels of PD-L1^{3RE} decreased faster than PD-L1^{WT}, therefore at 8 hr PD-L1^{3RE} level still decreased but it decreased in a much slower rate (Fig. R13G). We guess the decay of CHX is slower than the rate of PD-L1^{3RE} degradation, so we didn't see the recovery on PD-L1^{3RE} mutant.

Figure R13 Cellular levels of PD-L1^{WT} or PD-L1^{3RE} mutant. HEK293 cells expressing exogenous PD-L1^{WT} or PD-L1^{3RE} mutant were treated with 20 μ M cycloheximide (CHX) for 2, 4, 6, or 8 h. The PD-L1 level was analyzed by western blot (A-C). The intensities of the PD-L1 protein bands on the blots were quantified by ImageJ analysis, and the statistical data shown in (D) are the mean \pm SD from n = 3 independent experiments. ** P < 0.01 from two-sided Student's t-test.

4. Statistical analyses are properly conducted. But, please make sure what independent experiment means. Moreover, N=3 samples should be shown with dots.

Answer: Good suggestion! Statistical analyses have been double checked with the dots shown in the figures 4E and S8A.

REVIEWER COMMENTS

Reviewer #1 (Remarks to the Author):

At the outset, I respect and appreciate the quantum of work that the authors have put in. However, I have a fundamental problem with the central tenet of this paper. What is the relevance of the structure reported to the actual biology?

For example, Figure 2D shows a tight structure and Figure 2E shows that that is not the case. Even for the N-terminal residues, for example, residue 269, which according to the structure is well defined has a heteroNOE of 0.4 which indicates it is very dynamic. With only 11 long-range NOEs it is easy to get a tight bundle, that does not even look like a proper alpha-helix, with all the NOEs satisfied, that does not mean the structure is representative of what is actually in the membrane.

The free spectrum solution (without bicelles) is as well dispersed as the one in bicelles, Have the authors looked at the non-existence of the 11 long-range NOEs in the free solution sample.

Another major concern is regarding the biological relevance of their work, in particular with the lipid composition that was used in the structure determination, which was 100% POPG, arguably far from any biological lipid composition. And no proof is given that such a structure may occur at plasma membrane lipids. Some of the author's data in DMPG/DMPC suggest that CD260-290 binds membrane with increasing affinity with higher negatively charged compositions, but the data is not fully exploited. The authors should plot each titration point in terms of chemical shift perturbations and peak intensities, and conclude on how likely their structure is to happen at the plasma membrane. Furthermore, I still strongly suggest using a better membrane mimic such as 25-30% POPG – 70-75% POPC / DHPC bicelles.

The fact that Metformin prevents membrane association should be accompanied by heteroNOE data that shows that the protein does not bind to the membrane.

Reviewer #2 (Remarks to the Author):

The authors have responded to the reviewers' comments, including the presentation of considerable new experimental data. Though some questions remain for future evaluation, the current work will be of considerable interest and will serve as the basis for new mechanistic and potentially therapeutic opportunities. I believe this work is now appropriate for publication.

My only remaining comment would be for the authors to carefully consider the difference between "stability" and "degradation". On re-reading the manuscript, I get the impression that these terms are not being used with the care that is required. STABILITY typically refers to thermodynamic stability, whereas DEGRADATION may be related to stability, but formally refers to the observed half-life of a species. The authors might wish to make this distinction crystal clear.

Reviewer #3 (Remarks to the Author):

The authors have now included the details of the FRET experiment, which has made this reviewer understand what was done. The FRET efficiency can be translated to the distance between the fluorophores. I roughly calculated the distance between mTFP and rhodamine B in this experimental setup. The 75%, 50%, 10% FRET efficiency correspond to about 5, 6, and 8.5 nm, respectively. This information will be quite valuable for the researchers to figure out what happens in the cells.

I do not have further questions except for the following minor comments.

Line 625: To which dish or chamber were the one million cells seeded?

Line 626: 1640 must read as RPMI-1640.

REVIEWER COMMENTS

Reviewer #1 (Remarks to the Author):

At the outset, I respect and appreciate the quantum of work that the authors have put in. However, I have a fundamental problem with the central tenet of this paper. What is the relevance of the structure reported to the actual biology?

For example, Figure 2D shows a tight structure and Figure 2E shows that that is not the case. Even for the N-terminal residues, for example, residue 269, which according to the structure is well defined has a heteroNOE of 0.4 which indicates it is very dynamic. With only 11 long-range NOEs it is easy to get a tight bundle, that does not even look like a proper alpha-helix, with all the NOEs satisfied, that does not mean the structure is representative of what is actually in the membrane.

The free spectrum solution (without bicelles) is as well dispersed as the one in bicelles, Have the authors looked at the non-existence of the 11 long-range NOEs in the free solution sample.

Answer: The HetNOE values of the N-terminal 260-275 residues of CD₂₆₀₋₂₉₀ from the samples with bicelles are much higher than those from the samples without bicelles, indicating that CD₂₆₀₋₂₉₀ is an intrinsically disordered fragment of which the N-terminal 260-275 acquires partial structure upon interaction with lipids. However, even when bound to lipid bilayer, there is substantial ps dynamics as indicated by HetNOE data, which is consistent with the lack of rigid helical structure. The 11 NOEs mentioned above are long-range NOEs that we could detect. The structure was calculated to satisfy these NOEs but is by no means a rigid or static structure when bound to the bicelles; it is an over represented conformation reflected by our NMR data. We have removed the structure in the manuscript as suggested.

We have not collected the NOESY spectrum for CD₂₆₀₋₂₉₀ in solution (without bicelles) because we were not interested in the disordered structure in solution.

Another major concern is regarding the biological relevance of their work, in particular with the lipid composition that was used in the structure determination, which was 100% POPG, arguably far from any biological lipid composition. And no proof is given that such a structure may occur at plasma membrane lipids. Some of the author's data in DMPG/DMPC suggest that CD₂₆₀₋₂₉₀ binds membrane with increasing affinity with higher negatively charged compositions, but the data is not fully exploited. The authors should plot each titration point in terms of chemical shift perturbations and peak intensities, and conclude on how likely their structure is to happen at the plasma membrane. Furthermore, I still strongly suggest using a better membrane mimic such as 25-30% POPG – 70-75% POPE / DHPC bicelles.

Answer: We agree that 100% POPG is not a biological lipid composition. We used 100% POPG or 100% DMPG to trap CD₂₆₀₋₂₉₀ in a lipid-bound state to facilitate the characterization by NMR. To obtain further justification of this approach, we performed serial titrations of DMPG/DMPC ratio from 100% DMPC – 100% DMPG (see Figure S2). We found that the peaks in 260-275 moved in essentially straight lines, suggesting a two-state shift, from unbound to bound. Hence, at 25-30%

POPG, we expect the bound population will be lower but the conclusion is qualitatively the same. We have clarified the usage of 100% POPG or 100% DMPG in the main text. We further note that membrane anchoring of the CD₂₆₀₋₂₉₀ should also have the same effect of pushing the equilibrium to the bound state.

The fact that Metformin prevents membrane association should be accompanied by heteroNOE data that shows that the protein does not bind to the membrane.

Answer: We have performed the heteroNOE of CD₂₆₀₋₂₉₀ in the presence of metformin and observed similar dynamics as the protein in solution (Fig. S7), suggesting that metformin prevents membrane association, which is consistent with our conclusion.

Reviewer #2 (Remarks to the Author):

The authors have responded to the reviewers' comments, including the presentation of considerable new experimental data. Though some questions remain for future evaluation, the current work will be of considerable interest and will serve as the basis for new mechanistic and potentially therapeutic opportunities. I believe this work is now appropriate for publication.

My only remaining comment would be for the authors to carefully consider the difference between "stability" and "degradation". On re-reading the manuscript, I get the impression that these terms are not being used with the care that is required. STABILITY typically refers to thermodynamic stability, whereas DEGRADATION may be related to stability, but formally refers to the observed half-life of a species. The authors might wish to make this distinction crystal clear.

Answer: Good point! The palmitoylation of PD-L1 enhances the stability by adding a long chain lipid to PD-L1 which in turn inhibits ubiquitination and subsequent lysosomal degradation, while PD-L1 undergoes degradation in proteasomes or lysosomes with the involvement of SPOP or HIP1R in different pathways. We are not sure which pathway PD-L1-membrane interaction may contribute to, the palmitoylation enhanced stability, or the SPOP ubiquitination and HIP1R lysosomal degradation. It is possible that both stability and degradation are affected by the membrane association. In our manuscript, "stability" mainly is related to the palmitoylation, "degradation" mainly is related to SPOP and HIP1R degradation. We have carefully gone through the manuscript and confirmed or modified the usage of the two words "degradation" and "stability".

Reviewer #3 (Remarks to the Author):

The authors have now included the details of the FRET experiment, which has made this reviewer understand what was done. The FRET efficiency can be translated to the distance between the fluorophores. I roughly calculated the distance between mTFP and rhodamine B in this experimental setup. The 75%, 50%, 10% FRET efficiency correspond about 5, 6, and 8.5 nm,

respectively. This information will be quite valuable for the researchers to figure out what happens in the cells.

I do not have further questions except for the following minor comments.

Line 625: To which dish or chamber were the one million cells seeded?

Answer: In a 6-well plate.

Line 626: 1640 must read as RPMI-1640.

Answer: Corrected.

REVIEWERS' COMMENTS

Reviewer #1 (Remarks to the Author):

Since the structure is removed from the paper the NMR data largely support the claim and I recommend publication of this manuscript. As a continuing conversation with the authors, I would like to respond to their comments.

"The HetNOE values of the N-terminal 260-275 residues of CD260-290 from the samples with bicelles are much higher than those from the samples without bicelles, indicating that CD260-290 is an intrinsically disordered fragment of which the N-terminal 260-275 acquires partial structure upon interaction with lipids. "

This assessment is not true. the HetNOE data just informs that residues 260-275 are embedded in bicelles. The HetNOE data does not necessarily report on the structure.

"However, even when bound to lipid bilayer, there is substantial ps dynamics as indicated by HetNOE data, which is consistent with the lack of rigid helical structure."

Yet in the original manuscript, the RMSD of this part of the structure was tight and in conflict with the dynamics.

Since the structure is removed from the manuscript, these points are mute now.

Figure 2 still has the caption "The NMR structure of membrane-bound CD260-290"

Since there is no structure now, this caption must be changed.

REVIEWERS' COMMENTS

Reviewer #1 (Remarks to the Author):

Since the structure is removed from the paper the NMR data largely support the claim and I recommend publication of this manuscript. As a continuing conversation with the authors, I would like to respond to their comments.

"The HetNOE values of the N-terminal 260-275 residues of CD260-290 from the samples with bicelles are much higher than those from the samples without bicelles, indicating that CD260-290 is an intrinsically disordered fragment of which the N-terminal 260-275 acquires partial structure upon interaction with lipids. "

This assessment is not true. the HetNOE data just informs that residues 260-275 are embedded in bicelles. The HetNOE data does not necessarily report on the structure.

"However, even when bound to lipid bilayer, there is substantial ps dynamics as indicated by HetNOE data, which is consistent with the lack of rigid helical structure."

Yet in the original manuscript, the RMSD of this part of the structure was tight and in conflict with the dynamics.

Since the structure is removed from the manuscript, these points are mute now.

Answer: Thanks a lot for the thoughtful comments on the structure and dynamics, and we have greatly benefited from the discussion with the reviewer. We agree that the changes of the HetNOE values show the CD260-290 interaction with the bicelles, but not necessarily relate to

the conformational changes.

Figure 2 still has the caption "The NMR structure of membrane-bound CD260-290"

Since there is no structure now, this caption must be changed.

Answer: The caption has been changed to "The NMR characterization of membrane-bound CD260-290".

References:

1. Xu, C.; Gagnon, E.; Call, M.E.; Schnell, J.R.; Schwieters, C.D.; Carman, C.V.; Chou, J.J.; Wucherpennig, K.W. Regulation of t cell receptor activation by dynamic membrane binding of the cd3epsilon cytoplasmic tyrosine-based motif. *Cell* 2008, 135, 702-713.
2. Yang, W.; Pan, W.; Chen, S.; Trendel, N.; Jiang, S.; Xiao, F.; Xue, M.; Wu, W.; Peng, Z.; Li, X., et al. Dynamic regulation of cd28 conformation and signaling by charged lipids and ions. *Nat Struct Mol Biol* 2017, 24, 1081-1092.
3. Ulmer, T.S.; Bax, A.; Cole, N.B.; Nussbaum, R.L. Structure and dynamics of micelle-bound human alpha-synuclein. *J Biol Chem* 2005, 280, 9595-9603.
4. Han, C.; Shin, A.; Lee, J.; Lee, J.; Park, J.W.; Oh, J.H.; Kim, J. Dietary calcium intake and the risk of colorectal cancer: A case control study. *BMC cancer* 2015, 15, 966.
5. Wu, K.; Willett, W.C.; Fuchs, C.S.; Colditz, G.A.; Giovannucci, E.L. Calcium intake and risk of colon cancer in women and men. *Journal of the National Cancer Institute* 2002, 94, 437-446.
6. Wilson, K.M.; Shui, I.M.; Mucci, L.A.; Giovannucci, E. Calcium and phosphorus intake and prostate cancer risk: A 24-y follow-up study. *The American journal of clinical nutrition* 2015, 101, 173-183.
7. Zhang, T.; Muraih, J.K.; MacCormick, B.; Silverman, J.; Palmer, M. Daptomycin forms cation and size-selective pores in model membranes. *Biochimica et biophysica acta* 2014, 1838, 2425-2430.
8. Xue, J.; Li, L.; Li, N.; Li, F.; Qin, X.; Li, T.; Liu, M. Metformin suppresses cancer cell growth in

endometrial carcinoma by inhibiting pd-l1. *Eur J Pharmacol* 2019, 859, 172541.

9. Elgendy, M.; Ciro, M.; Hosseini, A.; Weiszmann, J.; Mazzarella, L.; Ferrari, E.; Cazzoli, R.; Curigliano, G.; DeCensi, A.; Bonanni, B., et al. Combination of hypoglycemia and metformin impairs tumor metabolic plasticity and growth by modulating the pp2a-gsk3beta-mcl-1 axis. *Cancer Cell* 2019, 35, 798-815 e795.

10. Cha, J.H.; Yang, W.H.; Xia, W.; Wei, Y.; Chan, L.C.; Lim, S.O.; Li, C.W.; Kim, T.; Chang, S.S.; Lee, H.H., et al. Metformin promotes antitumor immunity via endoplasmic-reticulum-associated degradation of pd-l1. *Mol Cell* 2018, 71, 606-620 e607.

11. Wang, Y.; An, H.; Liu, T.; Qin, C.; Sesaki, H.; Guo, S.; Radovick, S.; Hussain, M.; Maheshwari, A.; Wondisford, F.E., et al. Metformin improves mitochondrial respiratory activity through activation of ampk. *Cell Rep* 2019, 29, 1511-1523 e1515.

12. Lee, J.; Yesilkanal, A.E.; Wynne, J.P.; Frankenberger, C.; Liu, J.; Yan, J.; Elbaz, M.; Rabe, D.C.; Rustandy, F.D.; Tiwari, P., et al. Effective breast cancer combination therapy targeting bach1 and mitochondrial metabolism. *Nature* 2019, 568, 254-258.

13. DeFord-Watts, L.M.; Dougall, D.S.; Belkaya, S.; Johnson, B.A.; Eitson, J.L.; Roybal, K.T.; Barylko, B.; Albanesi, J.P.; Wulfing, C.; van Oers, N.S. The cd3 zeta subunit contains a phosphoinositide-binding motif that is required for the stable accumulation of tcr-cd3 complex at the immunological synapse. *J Immunol* 2011, 186, 6839-6847.

14. Zhang, H.; Cordoba, S.P.; Dushek, O.; van der Merwe, P.A. Basic residues in the t-cell receptor zeta cytoplasmic domain mediate membrane association and modulate signaling. *Proc Natl Acad Sci U S A* 2011, 108, 19323-19328.

15. Li, L.; Shi, X.; Guo, X.; Li, H.; Xu, C. Ionic protein-lipid interaction at the plasma membrane: What can the charge do? *Trends in biochemical sciences* 2014, 39, 130-140.

16. Okamoto, Y.; Bernstein, J.D.; Shikano, S. Role of c-terminal membrane-proximal basic residues in cell surface trafficking of hiv coreceptor gpr15 protein. *J Biol Chem* 2013, 288, 9189-9199.

17. Tetsuka, M.; Saito, Y.; Imai, K.; Doi, H.; Maruyama, K. The basic residues in the membrane-proximal c-terminal tail of the rat melanin-concentrating hormone receptor 1 are

- required for receptor function. *Endocrinology* 2004, 145, 3712-3723.
18. Lim, S.O.; Li, C.W.; Xia, W.; Cha, J.H.; Chan, L.C.; Wu, Y.; Chang, S.S.; Lin, W.C.; Hsu, J.M.; Hsu, Y.H., et al. Deubiquitination and stabilization of pd-l1 by csn5. *Cancer Cell* 2016, 30, 925-939.
19. Voli, F.; Valli, E.; Lerra, L.; Kimpton, K.; Saletta, F.; Giorgi, F.M.; Mercatelli, D.; Rouaen, J.R.C.; Shen, S.; Murray, J.E., et al. Intratumoral copper modulates pd-l1 expression and influences tumor immune evasion. *Cancer Res* 2020, 80, 4129-4144.
20. Yao, H.; Lan, J.; Li, C.; Shi, H.; Brosseau, J.P.; Wang, H.; Lu, H.; Fang, C.; Zhang, Y.; Liang, L., et al. Inhibiting pd-l1 palmitoylation enhances t-cell immune responses against tumours. *Nat Biomed Eng* 2019, 3, 306-317.
21. Rana, M.S.; Kumar, P.; Lee, C.J.; Verardi, R.; Rajashankar, K.R.; Banerjee, A. Fatty acyl recognition and transfer by an integral membrane s-acyltransferase. *Science* 2018, 359.
22. Zhang, J.; Bu, X.; Wang, H.; Zhu, Y.; Geng, Y.; Nihira, N.T.; Tan, Y.; Ci, Y.; Wu, F.; Dai, X., et al. Cyclin d-cdk4 kinase destabilizes pd-l1 via cullin 3-spop to control cancer immune surveillance. *Nature* 2018, 553, 91-95.
23. Kadkhodayan, S.; Sadat, S.M.; Irani, S.; Fotouhi, F.; Bolhassani, A. Generation of gfp native protein for detection of its intracellular uptake by cell-penetrating peptides. *Folia biologica* 2016, 62, 103-109.
24. Rozbicki, E.; Chuai, M.; Karjalainen, A.I.; Song, F.; Sang, H.M.; Martin, R.; Knolker, H.J.; MacDonald, M.P.; Weijer, C.J. Myosin-ii-mediated cell shape changes and cell intercalation contribute to primitive streak formation. *Nat Cell Biol* 2015, 17, 397-408.
25. Ferro, V.; Chuai, M.; McGloin, D.; Weijer, C.J. Measurement of junctional tension in epithelial cells at the onset of primitive streak formation in the chick embryo via non-destructive optical manipulation. *Development* 2020, 147.
26. Vial, G.; Demaille, D.; Guigas, B. Role of mitochondria in the mechanism(s) of action of metformin. *Frontiers in endocrinology* 2019, 10, 294.
27. Fontaine, E. Metformin-induced mitochondrial complex i inhibition: Facts, uncertainties, and consequences. *Frontiers in endocrinology* 2018, 9, 753.

28. Docrat, T.F.; Nagiah, S.; Naicker, N.; Baijnath, S.; Singh, S.; Chuturgoon, A.A. The protective effect of metformin on mitochondrial dysfunction and endoplasmic reticulum stress in diabetic mice brain. *Eur J Pharmacol* 2020, 875, 173059.